# A multiphase theory for spreading microbial swarms and films

**Siddarth Srinivasan[1], C Nadir Kaplan[1,2], L Mahadevan[1,2,3,4]\***

[1]John A. Paulson School of Engineering and Applied Sciences, Harvard University, Cambridge, United States; [2]Kavli Institute for Bionano Science and Technology, Harvard University, Cambridge, United States; [3]Department of Physics, Harvard University, Cambridge, United States; [4]Department of Organismic and Evolutionary Biology, Harvard University, Cambridge, United States

**Abstract** Bacterial swarming and biofilm formation are collective multicellular phenomena through which diverse microbial species colonize and spread over water-permeable tissue. During both modes of surface translocation, fluid uptake and transport play a key role in shaping the overall morphology and spreading dynamics. Here we develop a generalized two-phase thin-film model that couples bacterial growth, extracellular matrix swelling, fluid flow, and nutrient transport to describe the expansion of both highly motile bacterial swarms, and sessile bacterial biofilms. We show that swarm expansion corresponds to steady-state solutions in a nutrient-rich, capillarity dominated regime. In contrast, biofilm colony growth is described by transient solutions associated with a nutrient-limited, extracellular polymer stress driven limit. We apply our unified framework to explain a range of recent experimental observations of steady and unsteady expansion of microbial swarms and biofilms. Our results demonstrate how the physics of flow and transport in slender geometries serve to constrain biological organization in microbial communities.

DOI: https://doi.org/10.7554/eLife.42697.001

**\*For correspondence:**
lmahadev@g.harvard.edu

**Competing interests:** The authors declare that no competing interests exist.

## Introduction

Bacteria employ sophisticated surface translocation machinery to actively swarm, twitch, glide or slide over solid surfaces (*Kearns, 2010*; *Mattick, 2002*; *Spormann, 1999*; *Hölscher and Kovács, 2017*). Collectively, they also aggregate into multicellular communities on hydrated surfaces and exhibit large-scale coordinated movement (*Verstraeten et al., 2008*). Surface motility in macroscopic colonies on hydrated surfaces such as gels occurs primarily via two distinct modes: either by rapid flagella-mediated swarming expansion (*Harshey, 1994*; *Harshey, 2003*), or alternatively by slow biofilm expansion driven by extracellular polymer matrix production (*Hall-Stoodley et al., 2004*). In both cases, an interplay between mechanical constraints and biological organization sets limits on the overall colony morphology and expansion dynamics (*Persat et al., 2015*). The forces driving colony expansion are generated by non-homogeneous patterns of biological activity, originating from spatial localizations in cell growth and division (*Hamouche et al., 2017*), extracellular polymer matrix production (*Seminara et al., 2012*; *Yan et al., 2017*; *Srinivasan et al., 2018*), osmolyte secretion (*Ping et al., 2014*) and active stresses (*Farrell et al., 2013*; *Delarue et al., 2016*). Conversely, the formation of localized biologically active zones is tightly coupled to the heterogeneity of the environment, including the diffusion and transport of nutrients (*Wang et al., 2017*), accumulation of metabolic by-products (*Liu et al., 2015*; *Gozzi et al., 2017*) and presence of quorum sensing and signaling agents that regulate cell-differentiation and development.

Consequently, the dynamics of colony growth requires a mechanistic description that accounts for spatiotemporal inhomogeneities in biological activity, emergent forces, and flows that transport metabolic agents. In bacterial swarming, cells within the colony are actively propelled by the rotation

**eLife digest** Bacteria can grow and thrive in many different environments. Although we usually think of bacteria as single-celled organisms, they are not always solitary; they can also form groups containing large numbers of individuals. These aggregates work together as one super-colony, allowing the bacteria to feed and protect themselves more efficiently than they could as isolated cells.

These colonies move and grow in characteristic patterns as they respond to their environment. They can form swarms, like insects, or biofilms, which are thin, flat structures containing both cells and a film-like substance that the cells secrete. Availability of food and water influences the way colonies spread; however, since movement and growth are accompanied by mechanical forces, physical constraints are also important. These include the ability of the bacteria to change the water balance and their local mechanical environment, and the forces they create as they grow and move.

Previous research has used a variety of experimental and theoretical approaches to explain the dynamics of bacterial swarms and biofilms as separate phenomena. However, while they do differ biologically, they also share many physical characteristics.

Srinivasan et al. wanted to exploit these similarities, and use them to predict the growth and shape of biofilms and bacterial swarms under different conditions. To do this, a unified mathematical model for the growth of both swarms and biofilms was created. The model accounted for various factors, such as the transport of nutrients into the colony, the movement of water between the colony and the surface on which it grew, and mechanical changes in the environment (e.g. swelling/ softening). The theoretical results were then compared with results from experimental measurements of different bacterial aggregates grown on a soft, hydrated gel. For both swarms and biofilms, the model correctly predicted how fast the colony expanded overall, as well as the shape and location of actively growing regions.

Biofilms and other bacterial aggregates can cause diseases and increase inflammation in tissues, and also hinder industrial processes by damage to submerged surfaces, such as ships and waterpipes. The results described here may open up new approaches to restrict the spreading of bacterial aggregates by focusing on their physical constraints.

DOI: https://doi.org/10.7554/eLife.42697.002

of flagella in a thin layer of fluid extracted from the underlying soft tissue or gel (*Kearns, 2010*). In contrast, bacterial biofilms are surface aggregates of sessile bacteria embedded in a self-generated extracellular polymer matrix (*Flemming and Wingender, 2010*). Despite marked differences in regulatory genetic pathways, morphology and cell function (*Verstraeten et al., 2008*), physical characteristics such as the fluidization of the substrate/tissue, gradients in nutrient availability, the low-aspect-ratio geometry and the existence of multiple phases (i.e. cells, biopolymer and fluid) are common to both bacterial film and swarm colonies. Motivated by these similarities, we present a unified multi-phase framework that couples mechanics, hydrodynamics and transport to explain the dynamics of bacterial swarm and film expansion.

## Experimental background

### Bacterial swarms

Experiments on swarming colonies of *E. coli* (*Darnton et al., 2010*; *Wu and Berg, 2012*; *Ping et al., 2014*), *S. enterica* (*Harshey and Matsuyama, 1994*; *Butler et al., 2010*; *Kalai Chelvam et al., 2014*; *Chen et al., 2007*) and *P. aeruginosa* (*Yang et al., 2017*) reveal certain reproducible features associated with this modality of collective behavior. For example, *E. coli* swarms on agarose gels have a steady front shape that propagates radially at a uniform speed (*Wu and Berg, 2012*). In these swarms, measurements of the osmotic pressure profiles were found to be consistent with the active secretion of wetting agents in regions of high cell density that serve to fluidize the swarm by extracting water from the underlying tissue, thus allowing it to spread (*Ping et al., 2014*). These observations are not unique to *E. coli*; indeed our experiments with *B. subtilis* swarms, following (*Kearns and Losick, 2003*), indicate the same phenomena, that is a steady-state front shape and

speed, as shown in *Figure 1A–1E*. Close to the spreading front, we observe a multilayer region of width $W$ = 195 µm ± 35 µm, indicated by the dashed white lines in *Figure 1B and 1C*. The multilayer region correlates with increased colony thickness and local bacterial density (*Wu and Berg, 2012*). At the edge, and in the interior, there is just a monolayer of cells. The swarm radial expansion velocity is constant at $V$ = 2 mm/hr (see *Figure 1D*) and the swarm front maintains a steady-state profile during expansion (see *Figure 1E*). These observations raise a number of natural questions associated with the steady-state velocity and profile of the swarm colony. Given the observations of osmotic gradient-driven flow in the vicinity of the growing front (*Ping et al., 2014*), coupled with variations in the thickness and activity of bacteria, any framework to explain these requires a consideration of a dynamic bacterial population interacting with ambient fluid, necessitating a multiphase description.

## Bacterial films

In contrast with bacterial swarms, the spreading of bacterial biofilms is faciliated by the extracellular polymeric substance (EPS) matrix that expands via osmotic fluid influx, for example in *B. subtilis*

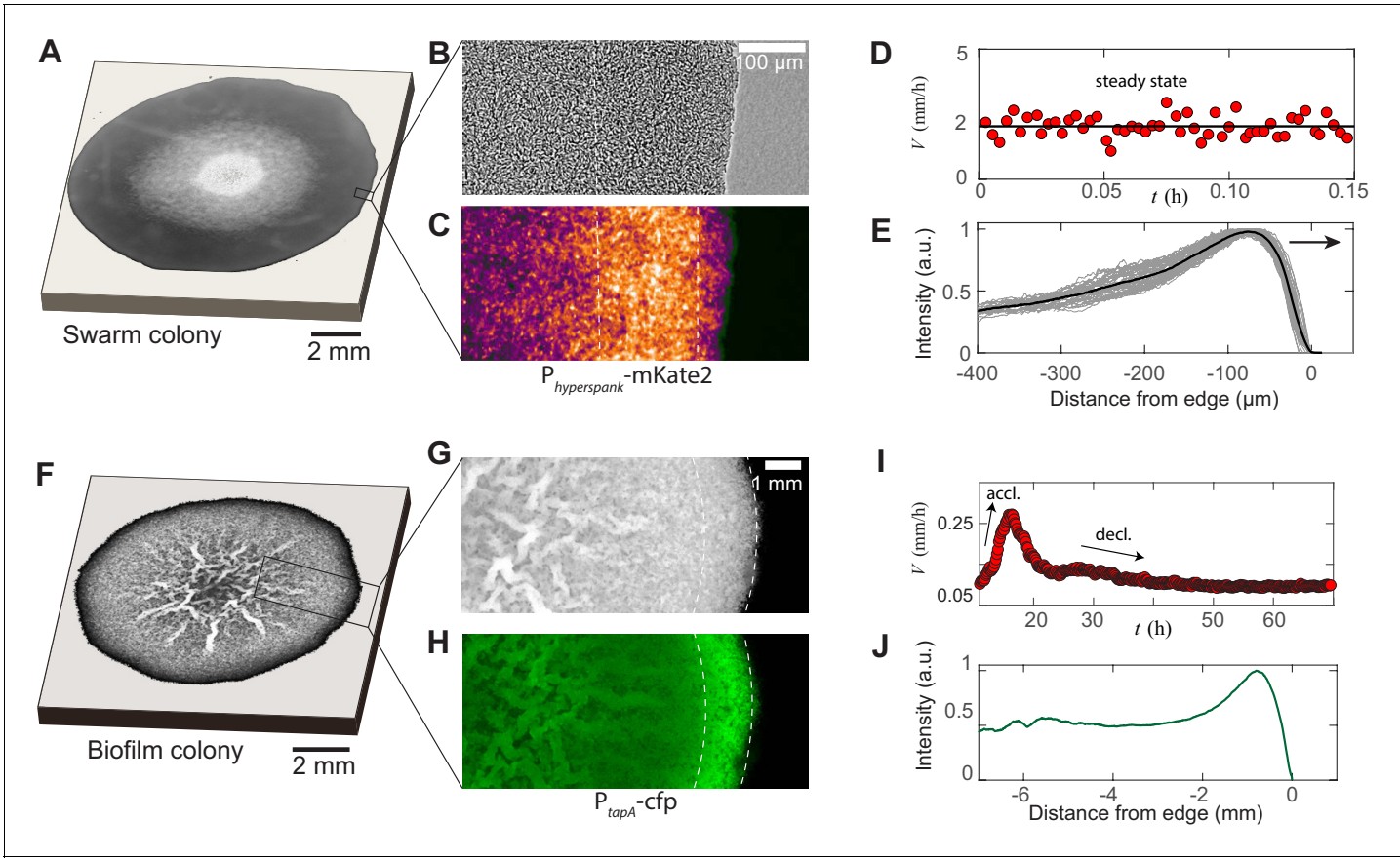

**Figure 1.** Experimental features of microbial swarms and biofilms. (A) Snapshot of a *Bacillus subtilis* swarm expanding on a 0.5 wt% LB/agar gel. (B,C) Brightfield and fluorescent zoom images of the leading swarm edge of a MTC822 strain containing the fluorescent $P_{hyperspank}$-mKate2 reporter that is expressed constitutively. The dashed white lines indicates the extent of the multi-cellular region. (D) Expansion velocity of the swarm measured at intervals of 10 s over a 10 min period. The solid line corresponds to a mean steady-state velocity of $V$ = 2 mm/h. (E) Mean intensity traces of the constitutive fluorophore (mKate2) representing bacterial densities profiles plotted in the moving steady-state frame. The dark grey traces represent separate density profile measurements taken every 10 s in the advancing swarm. The solid line represents the density profile averaged over a period of 30 min. (F) A *Bacillus subtilis* biofilm colony developing on a 1.5 wt% MSgg/agar gel. (G,H) Brightfield and fluorescent zoom images of the biofilm colony formed by a MTC832 strain harboring the $P_{tapA}$-cfp fluorescent reporter expressed in cells synthesizing the extracellular polymeric matrix (EPS). The dashed white lines indicates the extent of an active peripheral zone signifying localized EPS production. (I) Expansion velocity of the biofilm colony measured at intervals of 10 mins over a 72 hr period. The peak expansion velocity of $V$ = 0.22 mm/h occurs at $t \sim$ 18 h after inoculation. (J) Azimuthally averaged matrix reporter activity (cfp) as a function of spatial distance within the biofilm.
DOI: https://doi.org/10.7554/eLife.42697.003

(*Seminara et al., 2012*) and *V. cholerae* (*Yan et al., 2017*) biofilm colonies. However, EPS synthesis is not homogeneous, and depends on the local nutrient concentration and environmental heterogeneities experienced by cells within the same biofilm (*Vlamakis et al., 2008*; *Berk et al., 2012*). Recently, it was shown that the EPS matrix production is localized to cells in the propagating front of *B. subtilis* biofilms (*Srinivasan et al., 2018*). In *Figure 1F–1J*, we show the results of repeating these experiments, but now focusing on a peripheral region of a biofilm colony using a *B. subtilis* strain (MTC832) that harbors the $P_{tapA} - \mathrm{cfp}$ construct as a reporter for matrix production activity (*Wang et al., 2016*; *Srinivasan et al., 2018*). This highlights a $\sim 1$ mm zone of matrix production activity at the periphery, seen in *Figure 1G and H*; indeed plots of averaged matrix production reporter intensity exhibit a distinct peak at the periphery, as shown in *Figure 1J*. The dynamics of radial expansion shows the existence of an initial acceleration regime followed by a transition to a second regime characterized by a monotonic decrease in expansion velocity, as plotted in *Figure 1I*. This transient mode of biofilm spreading driven by EPS production and swelling is quite different from that of bacterial swarming, and suggests that we might need a fundamentally different way to address its origins. However, if we now consider the EPS matrix and fluid as distinct phases (*Cogan and Keener, 2004*; *Cogan and Keener, 2005*; *Winstanley et al., 2011*; *Seminara et al., 2012*), with the bacterial population being relatively small, we are again led to a multiphase description of the system, but with a different dominant balance relative to that seen in bacterial swarms, which we now turn to.

## Theoretical framework

Recent theoretical approaches have considered specific physical factors such as the wettability of the biofilm (*Trinschek et al., 2016*; *Trinschek et al., 2017*), osmotic pressure in the EPS matrix (*Winstanley et al., 2011*; *Seminara et al., 2012*), or Marangoni stresses associated with the swarm fluid (*Fauvart et al., 2012*), as reviewed by *Allen and Waclaw (2019)*. However, a description that captures the experimental observations described in *Figure 1* remains lacking. Here, given the similarities between the bacterial swarming and biofilm systems, we provide a unified description of their spreading dynamics by recognizing that in both cases we need to consider large slender microbial colonies with $H/R \ll 1$, where $H$ is the colony thickness and $R$ is the radius. This approximation results in a quasi-2-dimensional, two-phase model (assuming axisymmetry) of a colony that spreads along the x-axis, with a varying thickness, as shown in *Figure 2*. The subscript $i = (1,2)$ denotes the actively growing phase and passive phase, respectively. Within the swarm colonies, the highly motile cells constitute the actively growing phase whereas the fluid comprises the passive phase. Similarly, in biofilms, the EPS matrix constitutes the active phase, and the aqueous fluid is the passive phase.

In both cases, colony growth occurs over a semipermeable soft gel substrate, as shown in *Figure 2*. We develop a continuous description of colony expansion in terms of variables which are coarse-grained depth integrated averages (*Drew, 1983*; *Ishii and Hibiki, 2011*), The averaged height of the colony interface is $h(x,t)$, the volume fraction of the active phase (i.e., swarmer cells or polymer matrix) is $\phi_1 = \phi(x,t)$ and the volume fraction of the fluid phase is $\phi_2 = 1 - \phi(x,t)$. The 1-D substrate depth-averaged nutrient concentration field within the substrate is $c(x,t)$. As detailed in Appendix 2, combining mass and momentum balances yields the following generalized set of partial differential equations that governs the dynamics of both expanding swarms and biofilms,

$$((h\phi))_t + (Q_1(x))_x = g_1(h, \phi, c), \tag{1}$$

$$(h(1 - \phi))_t + (Q_2(x))_x = (1 - \phi)V_0(x), \tag{2}$$

$$c_t - Dc_{xx} = g_2(h, \phi, c). \tag{3}$$

where, $(\cdot)_x = \partial(\cdot)/\partial x$, etc. Here, $Q_1(x)$ is the horizontal flux in the active phase, $Q_2(x)$ is the horizontal flux in the fluid phase and $V_0(x)$ is the osmotically-driven net vertical fluid influx per unit length across the permeable substrate. Furthermore, $g_1(h, c, \phi)$ is the depth integrated active phase growth rate within the bacterial colony, and $g_2(h, c, \phi)$ is the depth integrated nutrient uptake rate. The dynamics of swarms and biofilms differ in the details of the expressions for $Q_1, Q_2, V_0$, which are provided in

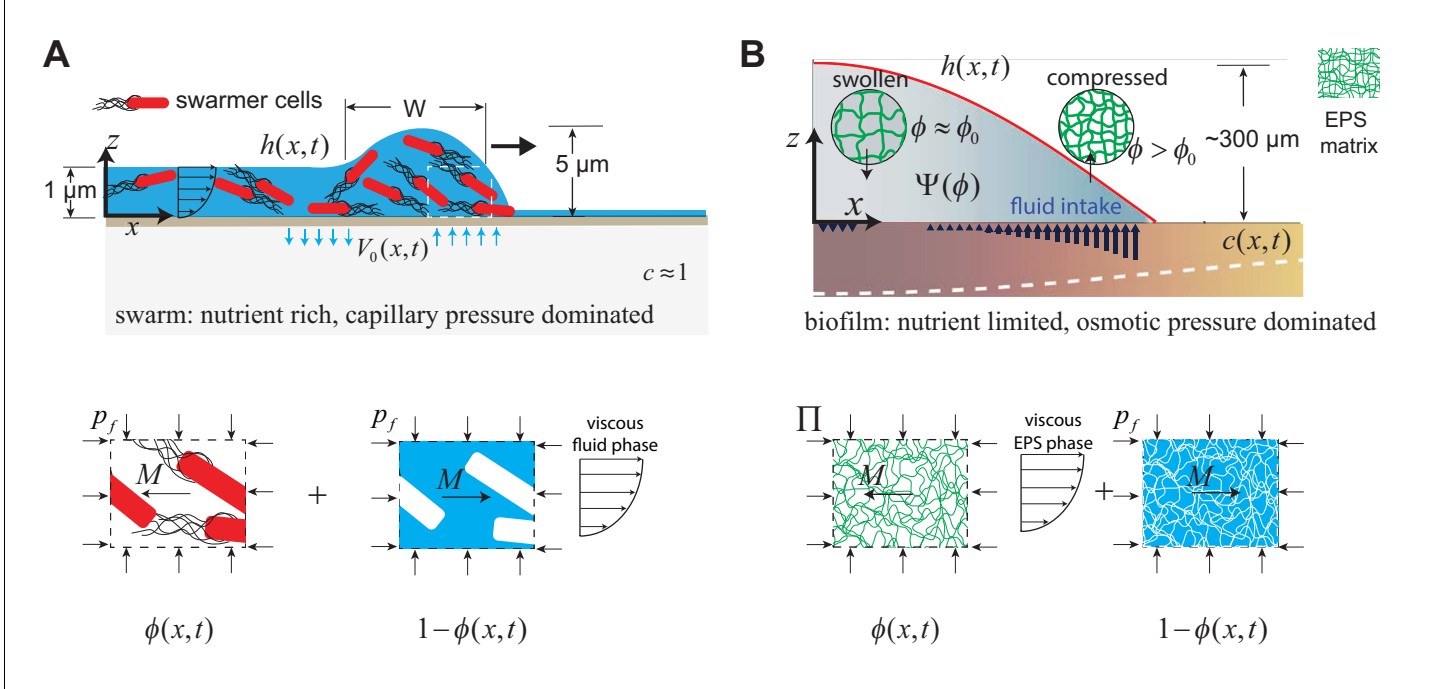

**Figure 2.** Geometry and variables governing colony expansion in (**A**) microbial swarms, and (**B**) bacterial biofilms, respectively. In both cases, the total thickness of the microbial colony is $h(x,t)$, the averaged nutrient concentration field is $c(x,t)$, the volume fraction of the active phase is $\phi(x,t)$, the volume fraction of the fluid phase is $1 - \phi(x,t)$, and the fluid influx across the agar/colony interface is denoted by $V_0(x,t)$. As shown on the bottom panel, the active phase constitutes swarmer cells in the microbial swarm, and secreted EPS polymer matrix in the biofilm. The pressure in the fluid phase is $p_f$ and the effective averaged pressure in the active phase is $\Pi$. In the swarm cell phase, $\Pi = p_f$, while the EPS phase effective pressure is $\Pi = p_f + \phi\Psi(\phi)$, where $\Psi(\phi)$ is the swelling pressure and is related to Flory-Huggins osmotic polymer stress (see **Equation A28**). The momentum exchange between the two phases is denoted by $M$, which includes the sum of an interfacial drag term and an interphase term as detailed in **Equation (A11)** in the Appendix.

DOI: https://doi.org/10.7554/eLife.42697.005

*Table 1.* While a full derivation of each term is provided in Appendix 2, a direct comparison of the

**Table 1.** Definitions of fluxes for swarms and films

Definitions of the active phase horizontal flux $Q_1$, the fluid phase horizontal flux $Q_2$, active phase growth term $g_1(h, \phi, c)$, osmotic influx term $V_0(x)$, and nutrient consumption term $g_2(h, \phi, c)$ for bacterial swarms and films in the generalized thin film evolution equations described by **Equations (1–3)**. Here, $\mu_1$ is the biofilm viscosity, $\mu_2$ is the fluid viscosity, $p_f$ is the fluid phase pressure, $\Pi$ is the effective pressure in the active phase, $g_0$ is effective swarmer cell growth rate, $G$ is the EPS production rate, $\Gamma$ is the nutrient consumption rate per unit concentration, $K$ is the nutrient half-velocity constant and $d$ is the thickness of the substrate. For swarms, the active phase corresponds to the swarmer cell phase, and for biofilms, the active phase is the EPS polymer matrix.

| | Variables | Swarms | Biofilms |
|---|---|---|---|
| **Flux (Phase I)** | $Q_1(x)$ | $-\frac{h^3}{3\mu_2}\frac{\phi}{1-\phi}\frac{\partial p_f}{\partial x} - \frac{h\phi}{\zeta}\frac{\partial p_f}{\partial x}$ | $-\frac{h^3}{3\mu_1}\frac{\partial \Pi}{\partial x}$ |
| **Flux (Phase II)** | $Q_2(x)$ | $-\frac{h^3}{3\mu_2}\frac{\partial p_f}{\partial x}$ | $-\frac{h^3}{3\mu_1}\frac{1-\phi}{\phi}\frac{\partial \Pi}{\partial x} - \frac{h}{\zeta}\frac{(1-\phi)^2}{\phi}\frac{\partial p_f}{\partial x}$ |
| **Osmotic influx** | $V_0(x)$ | $Q_0\left(\frac{\phi}{1-\phi} - \frac{\phi_0}{1-\phi_0}\right)$ | $Q_0\left(\phi^3 - \phi_0^3\right)$ |
| **Growth term** | $g_1(h, c, \phi)$ | $g_0 h\phi\left(1 - \frac{h\phi}{H\phi_0}\right)$ | $Gh\phi\frac{c}{K+c}\left(1 - \frac{h\phi}{H\phi_0}\right)$ |
| **Nutrient uptake** | $g_2(h, c, \phi)$ | - | $\frac{\Gamma\phi h}{d}\frac{c}{K+c}$ |

DOI: https://doi.org/10.7554/eLife.42697.004

terms listed in *Table 1* reveals a number of structural similarities and differences.

## Nutrient uptake

For both swarms and biofilms, the active phase (i.e., swarm cells or the EPS matrix) is generated within the bacterial colony by converting nutrient in the underlying substrate to biomass. The rate of change of nutrient concentration within the substrate depends on diffusion and nutrient uptake (see *Equation (3)* and *Equation (A2)*), and is derived in Appendix 2. When the substrate concentration is scaled by the initial concentration $c_0$, the nutrient depletion rate depends on $\Gamma/c_0$, the ratio of the specific nutrient consumption rate to the initial concentration. Bacterial swarming is typically associated with nutrient rich conditions, where $c_0 \gg \Gamma$. As a result, the nutrient uptake term can be neglected in bacterial swarming as $g_2 \to 0$, and the concentration $c \approx c_0$ throughout swarm expansion. In contrast, biofilm growth occurs under nutrient limited conditions where $\Gamma/c_0 \sim O(1)$, resulting in a corresponding uptake term shown in *Table 1*. Therefore, biofilm expansion is necessarily unsteady and driven by the dynamics of the transient nutrient field.

## Growth

In both swarms and biofilms, the generation of the active phase drives colony expansion and is described by the growth term in *Equation (1)* using a logistic function $g_1 = g_0 h\phi(1 - h\phi/(H\phi_0))$ to model the active phase growth, where $H\phi_0$ is the limiting thickness, and $g_0$ indicates a specific growth rate. In bacterial swarms, $g_0$ is independent of the nutrient concentration (as $c \approx c_0$ during swarm expansion). Therefore, the spreading swarm films have a steady-state structure that exhibits a central spatial plateau about $h\phi = H\phi_0$. In contrast, biofilm growth corresponds to a nutrient poor environment. We model the biofilm growth dependence on nutrient concentration via a minimal Michaelis-Menten form $g_0 = Gc/(K + c)$, . Unlike in nutrient rich conditions associated with swarms, this implies that biofilm growth is fundamentally transient; once the nutrient field at the interior is depleted as $c \to 0$, biofilm growth term in that region is arrested and $g_1 \to 0$ independently of the vertical thickness (i.e., even if $h\phi \neq H\phi_0$). As a result, the biofilm does not give form a central plateau and the dynamics of the biofilm rim is fixed by the dynamics of nutrient depletion. Eventually the effect of the finite-size of the system (the petri dish) also becomes important it determines the overall dynamics of nutrient depletion.

## Active and passive fluxes

The terms $Q_1(x)$ and $Q_2(x)$ that represent the horizontal flux of the active and passive phases are obtained by depth integrating the momentum balance equations in the thin-film lubrication limit, as described in Appendix 2 (c.f. *Equations (A9)-(A11)*). Within bacterial swarms, the passive aqueous fluid phase is modeled as a Newtonian liquid with viscosity $\mu_2$. The first term of $Q_1(x)$ and $Q_2(x)$ in *Table 1* for swarms is generated by viscous and capillary stresses within the swarm fluid. The active swarmer cells are treated as inviscid and subjected to a hydrodynamic frictional drag force. Specifically, we assume that individual bacteria within the swarm are undergoing a random walk process with zero net displacement (upon averaging over sufficiently large time-intervals). Even though there is no overall displacement, there is a net time-averaged drift that arises from viscous stokes drag interaction between the fluid and the active bacteria. The second term for $Q_1(x)$ in *Table 1* represents this time-averaged drift arising from frictional drag interaction of the bacteria with the swarm fluid.

In biofilms, the EPS matrix phase constitutes an active viscous hydrogel network with viscosity $\mu_1$, whereas the passive aqueous fluid phase is treated as a solvent with viscosity $\mu_2$. The dominant stress within the EPS phase in the biofilm model arises from a Flory-Huggins swelling pressure in the polymer chains (*Cogan and Guy, 2010*; *Winstanley et al., 2011*). In the fluid phase, the pressure $p_f$ is set by surface tension and curvature of the swarm fluid. Both these stresses contribute to the effective EPS phase pressure term $\Pi(x)$, as described in Appendix 2. Consequently, the first term for $Q_1(x)$ and $Q_2(x)$ in *Table 1* for biofilms is related to the gradient of the effective pressure. Moreover, following *Winstanley et al. (2011)*, we assume that the capillary and viscous stresses in the swarm fluid are negligible when compared to the frictional drag due to flow between water and the EPS polymer chain network in the biofilm model. Therefore, the second term for $Q_2(x)$ in *Table 1* represents a Darcy-type flow of the aqueous phase within the EPS matrix. The osmotic influx terms are considered separately in the following sections when describing the equations governing swarm and biofilm expansion.

## Bacterial swarms

Species of bacteria that swarm on hydrated surfaces are known to secrete distinct wetting agents. For example, *B. subtilis* secretes the lipopeptide surfactin, whereas *P. aeruginosa* secrets rhamnolipids as the wetting agent. Consequently, existing thin-film models to describe bacterial swarming assume that gradients in wetting agent activity generate Marangoni stresses that drives swarming motility (*Fauvart et al., 2012*; *Trinschek et al., 2018*). However, *E. coli* exhibits swarming behavior despite the absence of lipopeptides or other agents that act as surfactants. Moreover, recent experiments (*Yang et al., 2017*) demonstrates that *P. aeruginosa* swarms robustly even after exogenously eliminating gradients in surfactant concentration within the swarm fluid, eliminating Marangoni flows as the principal mechanism that drives swarming. Here, we take a different approach based on experiments that show that steady-state swarm colony expansion maybe mediated by secretion of agents that are osmotically active (*Wu and Berg, 2012*). As we will see, this leads to fluid being extracted from the substrate near the front, then driven into the colony by capillary and viscous stresses, and eventually returns into the substrate in the interior of the swarm.

Within the bacterial swarms, the dominant phases are the swarmer cell phase, and the viscous aqueous phase, as shown in the bottom panel of *Figure 2A*. Fluid uptake from the substrate is regulated by the secretion of osmotically active agents by the swarmer cells (*Ping et al., 2014*). We represent the osmotic agent in the fluid by a concentration field, $c_{\text{osm}}(\phi)$ that is proportional to the local volume fraction of cells such that $c \propto \phi/(1-\phi)$, and gives rise to an osmotic pressure described by van't Hoff's law as (*van't Hoff, 1887*), $\Delta\Psi = \left(\Psi_0\phi/(1-\phi) - \Psi_{\text{eq}}\right)$, that drives the fluid intake. Here, $\Psi_0$ is the osmotic pressure scale in the swarm fluid and $\Psi_{\text{eq}}$ is the equilibrium osmotic pressure within the underlying tissue/gel substrate. Away from the front, in the interior of the swarm colony, there is no net fluid influx (*Ping et al., 2014*). Therefore, the equilibirium volume fraction of the swarm cells at the interior is, $\phi_0 = \Psi_{\text{eq}}/(\Psi_0 + \Psi_{\text{eq}})$. At the front itself, the difference in osmotic pressure results in a net Darcy-type fluid influx into the swarm, $V_0(x)$, expressed as,

$$V_0(x) = Q_0\left(\frac{\phi(x)}{1-\phi(x)} - \frac{\phi_0}{1-\phi_0}\right). \tag{4}$$

where $Q_0$ is a velocity scale associated with fluid inflow from the substrate. Measurements of cell replication within swarms reveals that growth is restricted to swarmer cells at the periphery (*Hamouche et al., 2017*), which we model using a modified logistic growth term $g_1(h, \phi)$ as listed in *Table 1*, that localizes all cell division to the periphery. Here, $H\phi_0$ is the limiting thickness of the swarmer-cell phase at the interior, and $g_0$ is an effective specific growth rate, related to true specific cell growth rate by a geometric factor (see discussion in Appendix. [2]).

## Parameters and scaling laws for bacterial swarms

To make sense of the scales in the problem, we use the dimensionless variables $\hat{x} = x/L$, $\hat{z} = z/H$ and $\hat{t} = tg_0$ where H is the vertical length scale, L is a horizontal length scale and $1/g_0$ is the time-scale associated with bacterial growth. The resultant horizontal velocity scale in the swarm colony is $U = Lg_0$. Swarm expansion is fluid driven, and therefore balancing the viscous stresses generated in the swarm fluid, with the curvature pressure due to surface tension (*Levich and Landau, 1942*) results in $\mu_2 U/H^2 \sim \gamma H/L^3$, where $\mu_2$ is the viscosity and $\gamma$ is the surface tension of the aqueous phase. As a result, the natural horizontal length scale is $L = H(Ca)^{-1/3}$, where $Ca = (\mu_2 U/\gamma)$ is a capillary number associated with the microbial swarm fluid. Consequently, in our model the expansion speed of the swarm colony, $V = dR/dt$, is determined by the product of the horizontal length scale and an effective growth rate, and is predicted to scale as,

$$V = C_1 g_0 H Ca^{-1/3}. \tag{5}$$

whereas, the swarm front itself is analogous to a capillary ridge in thin fluid film with a width $W$ that is predicted to scale as,

$$W = C_2 Ca^{-1/3}. \tag{6}$$

where, $C_1$ and $C_2$ are dimensionless prefactors that require a detailed numerical calculation, and are

discussed later. There are two important dimensionless parameters that describe swarm colony expansion. The first dimensionless parameter, $\alpha_1$, relates the magnitude of capillary forces to the viscous drag acting on cells within the swarm and is defined as $\alpha_1 = (\gamma H/L^2)/(\zeta L U)$. Here, $\zeta = \zeta_c/V_c$ where $\zeta_c$ is the friction coefficient of a single swarmer cell and $V_c$ is its volume. The second dimensionless parameter $\alpha_2$ is defined as the ratio of a vertical fluid influx velocity $Q_0$, to a thickness velocity scale $Hg_0$ associated with bacterial growth as $\alpha_2 = Q_0/(Hg_0)$.

The vertical length scale and equilibrium fluid volume fraction are estimated from the interior monolayer region as $H = 0.5\,\mu\text{m}$ and $\phi_0 = 0.5$ (*Wu and Berg, 2012*). We assume values of $\mu_2 = 10^{-3}$ Pa.s for the (aqueous) swarm fluid viscosity, and $\gamma = 10^{-2}$ N/m as its surface tension. The friction coefficient of a single cell is estimated from Stokes law as $\zeta_c = 3\pi\mu_2 a$, and its volume is approximated as $V_c = \pi a^3/6$, where $a = 1\,\mu\text{m}$ is the cell diameter. Therefore, the friction coefficient is $\zeta = \zeta_c/V_c \approx 18\mu_2/a^2$. As a result of substituting the values of known parameters above, the dimensionless parameter $\alpha_1$ reduces to a constant geometric ratio, $\alpha_1 \approx 2a^2/H^2 \approx 2/9 \approx 0.22$.

The value of $\alpha_2$ depends on the ratio $Q_0/g_0$. Direct experimental measurements of the vertical influx fluid velocity profile $V_0(x)$ and the spatial profiles of cell division in swarm colonies remain scarce (*Hamouche et al., 2017*). In order to make progress in validating our model with real experimental data, the vertical fluid influx velocity scale is chosen as $Q_0 = 10^{-2}\,\mu\text{m/s}$. Consequently, we have chosen $g_0$ as the only fitting parameter in our study, as detailed in Appendix 2. As an example, in the following section we will show that a choice of $g_0 = 0.013\ \text{s}^{-1}$ in our model reproduces the experimental swarm expansion speed shown in *Figure 1D*, and leads to a horizontal length scale of $L = H(Ca)^{-1/3} = 100\,\mu\text{m}$, velocity scale of $U = Lg_0 = 1.3\,\mu\text{m s}^{-1}$, $Ca = 1.3 \times 10^{-7}$ and a value of $\alpha_2 \approx 1.5$. A complete set of parameters for three experimental measurements of swarm expansion in *B. subtilis*, and two existing measurements in *E. coli* previously reported by *Darnton et al. (2010)* and *Wu and Berg (2012)* are summarized in Appendix 2.

## Steady state swarms

With these assumptions, and assuming that the nutrient concentration is constant, *Equations (1–3)* reduce to the following scaled equations in the swarming limit,

$$(\hat{h}\phi)_{\hat{t}} + \frac{1}{3}\left(\frac{\phi\hat{h}^3\hat{h}_{\hat{x}\hat{x}\hat{x}}}{1-\phi}\right)_{\hat{x}} + \alpha_1\left(\hat{h}\hat{h}_{\hat{x}\hat{x}\hat{x}}\phi\right)_{\hat{x}} = \hat{h}\phi(1-\hat{h}\phi) \tag{7}$$

$$\left(\hat{h}(1-\phi)\right)_{\hat{t}} + \frac{1}{3}\left(\hat{h}^3\hat{h}_{\hat{x}\hat{x}\hat{x}}\right)_{\hat{x}} = \alpha_2\frac{\phi - \phi_0}{1-\phi_0}. \tag{8}$$

To complete the formulation of the problem, we need five boundary conditions which are $\hat{h}_{\hat{x}}(0) = \hat{h}_{\hat{x}}(R_P) = 0$, $\hat{h}_{\hat{x}\hat{x}\hat{x}}(0) = \hat{h}_{\hat{x}\hat{x}\hat{x}}(R_P) = 0$, and $\phi(0) = \phi_0$, where $R_P$ is the dimensionless size of the petri-dish and is set much larger than the colony size ($\hat{R} = 150$) in our simulations. The initial condition corresponds to a circularly inoculated swarm colony, along with a thin pre-wetting film where no bacterial growth occurs (see *Appendix 3—figure 1*).

Solving *Equations (7) and (8)* with the prescribed initial and boundary conditions numerically results in a steady state solution that advances at a constant speed (see *Figure 3*). In *Figure 3*, we plot a representative steady state solution in the frame of the advancing front for $\alpha_1 = 0.2$, $\alpha_2 = 1.5$ and $\phi_0 = 0.5$. At the interior of the swarm, the average cell volume fraction is $\phi \approx \phi_0$. Near the leading edge of the swarm, there is a region of enhanced thickness as indicated by the red line in *Figure 3A*. Immediately behind the leading edge, where the cell concentration is highest, so is the osmolyte concentration leading to fluid extraction from the substrate, while further behind, fluid is reabsorbed, as indicated by the arrows in *Figure 3A*. In *Figure 3B*, we show the steady-state osmotic flow solution and see that it correlates well with the experimentally measured osmotic pressure profile by *Ping et al. (2014)* in *E. coli* swarms. As shown in *Appendix 3—figure 3*, our numerical horizontal flow profiles are also consistent the scaled radial fluid velocity measurements of *Wu and Berg (2012)*. In *Figure 3C*, we see that the radial expansion velocity scales as $Hg_0$ and shows quantitative agreement with experiments and is insensitive to the fluid influx velocity scale when $Q_0 \gg g_0H$. Note that our model uses a coarse-graining procedure and represents the swarm thickness field using a continuum approximation. As a consequence, we are not able to

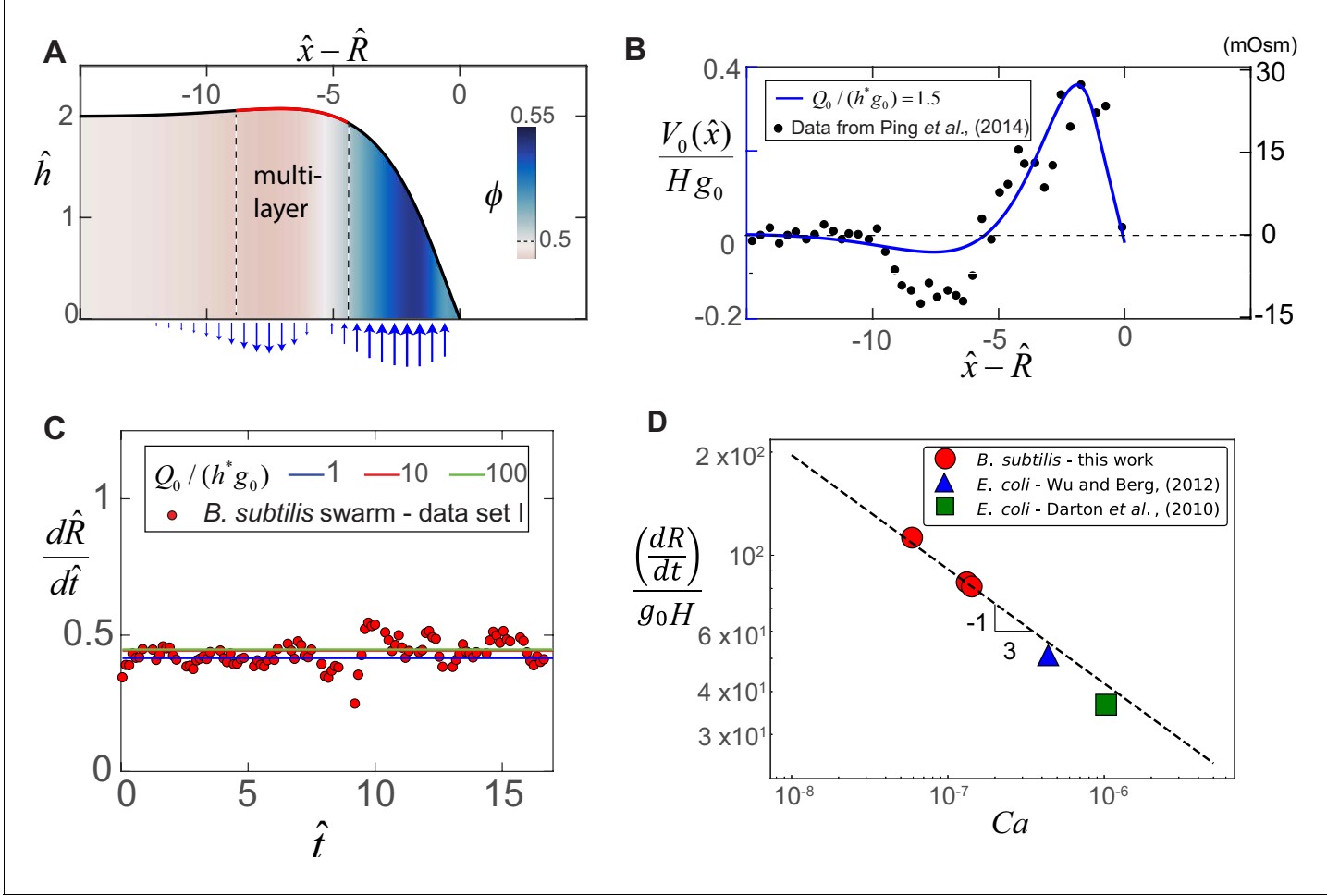

**Figure 3.** Steady-state morphology and fluid transport in a bacterial swarm obtained by solving *(8)* for $\alpha_1 = 0.2$, $\alpha_2 = 1.5$ and $\phi_0 = 0.5$. (A) Plot of the steady-state thickness $\hat{h} = h/H$ against the scaled distance $\hat{x} - \hat{R}$, where $\hat{x} = x/(HCa^{-1/3})$ and $\hat{R}$ is the radius. The solid red line indicates a region of increased thickness, and the colormap quantifies variations in $\phi$, the local volume fraction. (B) Plotted on the left-axis is the numerical steady-state fluid uptake profile within the swarm (solid line) calculated from *Equation (4)*. On the right axis are experimental measurements of the steady-state osmotic pressure within an expanding *E. coli* swarm (filled circles), reproduced from *Ping et al. (2014)*, with the baseline reference value shifted to zero, and with distances normalized by $L = 50$ μm. (C) Predicted steady-state radial colony expansion speeds within the swarm for values of $\alpha_2 = Q_0/(Hg_0) = 1$, 10 and 100 respectively. The data points are expansion speeds in *B. subtilis* swarms measured over 20 min, and scaled using $U = 1.3$ μms$^{-1}$ and $g_0 0.013$ s$^{-1}$. (D) Comparison between the swarm expansion velocities $dR/dt$ measured for five separate colonies (see Appendix 2) and the estimated capillary number. For each experiment, $g_0$ was obtained by fitting the steady state solution of *Equations (7) and (8)* to the swarm velocity. The dashed line corresponds to the predicted scaling law in *Equation (5)*.

DOI: https://doi.org/10.7554/eLife.42697.006

quantitatively capture the decreasing height of the swarm (i.e., of the order of a few cells), that is experimentally observed over hundreds of micron towards the interior (see *Figure 1E*).

Furthermore, we corroborate our scaling law in *Equation (5)* by fitting our model to five independent experimental measurements of swarm expansion velocities for different systems, as shown in *Figure 3D*. These include measurements in *B. subtilis* swarms in this work, and in *E. coli* swarms previously reported by *Darnton et al. (2010)* and *Wu and Berg (2012)* that are summarized in Table A2 in Appendix 2. The expansion velocity follows the $-1/3$ exponent predicted by *Equation (5)* for $Ca$ varying from $\sim 5 \times 10^{-8}$ to $10^{-6}$. For each experiment, we have fit our theoretical model using the effective growth rate $g_0$ as the fitting parameter and find that the numerical prefactor $C_1 \approx 0.42$. However, as shown in *Appendix 3—figure 5* in the Appendix, the measured multi-layer width does not follow the predicted scaling. From an experimental point of view, the width of the multi-layer region is not sharply defined in *Figure 1E*, and will depend on the choice of threshold. However, our multi-

phase model is able to describe the zone of cellular and osmolyte activity near the leading edge that drives the advancing swarm front. This leads to a picture wherein the combination of a fluid-filled substrate and swarm front work together like a localized active circulatory system, quantitatively rationalizing the experimental observations of *Wu and Berg (2012)* and *Ping et al. (2014)*.

## Bacterial films

In bacterial biofilms, the EPS matrix secreted by bacteria constitutes the active phase and undergoes swelling, drawing in the fluid that acts as the passive phase. As shown in *Figure 2B*, the EPS is initially synthesized in a partially swollen, out-of-equilibrium state at the periphery. The polymer chains gradually relax to an equilibrium fully-swollen configuration by the generation of a swelling pressure $\Psi$ within the biofilm, and via fluid uptake $V_0(x)$ from the substrate. As discussed in Appendix 2, the swelling pressure is $\Psi(\phi) = \psi(\phi)/\phi$, where $\psi(\phi) = \psi_0 \times \phi^3$ is the osmotic pressure in the EPS matrix using the Flory-Huggins model for a polymer network in a $\theta$-solvent (*Rubinstein and Colby, 2003*), where $\psi_0 = kT/(b^3)$ is the osmotic pressure scale, $kT$ is the product of the Boltzmann constant with the temperature and $b$ is the approximate size of the monomer unit. The net effective pressure term driving biofilm expansion is, $\Pi = \psi_0 \phi^3 + p_f$, where $p_f$ is the capillary pressure, so that the water influx across the substrate is

$$V_0(x) = Q_0(\phi^3 - \phi_0^3). \tag{9}$$

where $Q_0$ is the influx fluid velocity scale, $\phi_0 = (\Psi_{\text{eq}}/\Psi_0)^{1/3}$ is the fully-swollen EPS polymer volume fraction and $\Psi_{\text{eq}}$ is the osmotic pressure of the substrate over which the colony grows. Finally, nutrient uptake is modeled by a Monod growth law, while the synthesis of the EPS matrix is modeled by a logistic term as listed in *Table 1*.

### Parameters and scaling estimates for bacterial films

We consider dimensionless variables $\hat{x} = x/L$, $\hat{z} = z/H$, $\hat{t} = tG$, $\hat{\phi} = \phi/\phi_0$ and $\hat{c} = c/c_0$, where $H$ is now the maximum biofilm thickness, $G$ is the rate of EPS production, and $c_0$ is the initial nutrient concentration in the substrate. As biofilm growth is nutrient limited (*Liu et al., 2015*), the dimensionless length scale is determined from *Equation (3)* and is expected to scale as $L = (D/G)^{1/2}$ and the corresponding velocity scale is $U = (DG)^{1/2}$.

Using these scales, we can define the ratio of osmotic stresses relative to viscous stress in the EPS phase in terms of the dimensionless parameter, $\beta_1 = (\Psi_0/L)/(\mu_1 U/H^2)$, the ratio of capillary stresses relative to the EPS viscous stress in terms of another parameter, $\beta_2 = (\gamma H/L^3)/(\mu_1 U/H^2)$, the ratio of capillary stress to the interfacial drag in the aqueous fluid phase, $\beta_3 = (\gamma H/L^2)/(\zeta U L)$, and the ratio of the fluid influx velocity to the EPS swelling velocity, $\beta_4 = Q_0/(HG)$. As shown in Appendix 2, the effective nutrient uptake rate is $S = (\Gamma H \phi_0)/(c_0 d)$, where $\Gamma$ is the nutrient consumption rate per unit concentration and $d$ is the substrate thickness. Consequently, we define $\beta_5 = S/G$ as the ratio of the effective nutrient uptake rate to the EPS production rate.

We set the EPS production time-scale as $G = 1/40$ min$^{-1}$, resulting in a horizontal length scale of $L = (D/G)^{1/2} = 1.1$ mm and velocity scale $U = (DG)^{1/2} = 0.5$ $\mu$m/s. The effective nutrient uptake rate is estimated as $S = 1/25$ min$^{-1}$, where we have taken $d = 7$ mm as the substrate thickness (*Srinivasan et al., 2018*), $\Gamma = 10^{-2}$ mM/s as the nutrient uptake rate (*Zhang et al., 2010*), and $c_0 = 35$ mM as the initial concentration of the carbon source. The friction coefficient is $\zeta \sim \mu_2/\xi^2$, where the EPS mesh size is $\xi = 50$ nm (*Yan et al., 2017*). Using measured estimates of the biofilm viscosity $\mu_1 = 10^5$ Pa.s (*Stoodley et al., 2002*; *Lau et al., 2009*), fluid phase viscosity $\mu_2 = 10^{-3}$ Pa.s, surface tension $\gamma = 10^{-2}$ N/m, an osmotic scale $\Psi_0 = 2100$ Pa (*Yan et al., 2017*) (i.e., $\phi_0 = 0.04$), biofilm thickness $H = 400 \mu$m, and nutrient diffusivity in agarose gels of $D = 5 \times 10^{-10}$/s (*Zhang et al., 2010*) implies that $\beta_1 \approx 7$, $\beta_2 \approx 0.01$, $\beta_3 \approx 0.02$, $\beta_4 \approx 1$ and $\beta_5 \approx 2$. Consequently, within the context of our model, it is evident that osmotic stresses, fluid influx and biomass growth are the dominant forces that drive colony expansion. Moreover, in the nutrient limited regime, our model predicts the transient maximum biofilm expansion velocity to scale as,

$$V = C_3 (DG)^{\frac{1}{2}} \tag{10}$$

whereas, the width of the propagating fronts of EPS production experimentally observed by *Srinivasan et al. (2018)* is predicted to scale according to,

$$W = C_4 \left(\frac{D}{G}\right)^{\frac{1}{2}} \tag{11}$$

where $C_3$ and $C_4$ are once again dimensionless prefactors that require a detailed numerical calculation, as discussed later.

## Transient biofilm solutions

With the above scaling assumptions, *Equations (1–3)* now reduces to the following partial differential equations that describe biofilm colony expansion,

$$\left(\hat{h}\hat{\phi}\right)_{\hat{t}} - \frac{\beta_1}{3\phi_0}\left(\hat{h}^3(\hat{\phi}^3)_{\hat{x}}\right)_{\hat{x}} + \frac{\beta_2}{3\phi_0}\left(\hat{h}^3\hat{h}_{\hat{x}\hat{x}\hat{x}}\right)_{\hat{x}} = \frac{\hat{c}\hat{h}\hat{\phi}(1-\hat{h}\hat{\phi})}{K_1+\hat{c}}, \tag{12}$$

$$\begin{aligned}
&\left(\hat{h}\left(1-\phi_0\hat{\phi}\right)\right)_{\hat{t}} - \frac{\beta_1}{3}\left(\kappa(\hat{\phi})\hat{h}^3(\hat{\phi}^3)_{\hat{x}}\right)_{\hat{x}} \\
&+ \frac{\beta_2}{3}\left(\kappa(\hat{\phi})\hat{h}^3\hat{h}_{\hat{x}\hat{x}\hat{x}}\right)_{\hat{x}} + \beta_3\left(\hat{h}(1-\phi_0\hat{\phi})\kappa(\hat{\phi})\hat{h}_{\hat{x}\hat{x}\hat{x}}\right)_{\hat{x}} \\
&= \beta_4(1-\phi_0\hat{\phi})\left(\hat{\phi}^3-1\right),
\end{aligned} \tag{13}$$

$$\hat{c}_{\hat{t}} - \hat{c}_{\hat{x}\hat{x}} = -\beta_5\frac{\hat{h}\hat{\phi}\hat{c}}{K_1+\hat{c}}. \tag{14}$$

where $\kappa(\hat{\phi}) = (1-\phi_0\hat{\phi})/(\phi_0\hat{\phi})$ is a volume fraction dependent permeability term. The eight boundary conditions associated with *Equations (12)–(14)* are the symmetry boundary conditions $\hat{h}_{\hat{x}}(0) = \hat{h}_{\hat{x}}(R_P) = 0$, $\hat{h}_{\hat{x}\hat{x}\hat{x}}(0) = \hat{h}_{\hat{x}\hat{x}\hat{x}}(R_P) = 0$, $\hat{\phi}_{\hat{x}}(0) = \hat{\phi}_{\hat{x}}(R_P) = 0$, $\hat{\phi}_{\hat{x}\hat{x}\hat{x}}(0) = \hat{\phi}_{\hat{x}\hat{x}\hat{x}}(R_P) = 0$ and $\hat{c}_{\hat{x}}(0) = \hat{c}_{\hat{x}}(R_P) = 0$, where the dimensionless petri-dish size is chosen as $R_P = 16$ to match the size of typical 35 mm diameter petri dishes used in experiments (*Srinivasan et al., 2018*). In *Figure 4A*, we plot the time evolution of the shape and nutrient concentration field for a biofilm colony of initial radius $\hat{R}_0 = 0.5$ and thickness $\hat{h}_{\text{in}} = 0.06$.

Unlike in the case of swarms, the solutions to *Equations (12)–(14)* are transient, and exhibit two distinct expansion regimes: initial acceleration phase until $\hat{t}_c = 5$, followed by a decelerating phase beyond. For $\hat{t} < \hat{t}_c$, colony expansion arises as the microbes rapidly consumes locally available nutrient at the interior and synthesize fresh EPS matrix, generating spatial gradients in nutrient availability (see *Figure 4A*). In *Figure 4B*, we show that the newly synthesized EPS generates a large osmotic pressure differential between the biofilm and the substrate, and osmotic fluid influx gradually relaxes the biofilm matrix to a swollen configuration. For $t > t_c$, the localized zone of EPS production near the film front propagates with a fixed shape as shown in *Figure 4C*, consistent with the observed spatial localization in *tapA* gene activity (see *Figure 1J* and *Srinivasan et al., 2018*). Moreover, the radial colony expansion profile in *Figure 4D* is also consistent with the non-monotonic front speed observed experimentally (*Srinivasan et al., 2018*). For the specific experimental conditions we consider, our detailed theory allows us to estimate the prefactors in the scaling laws *Equations (10)– (11)* so that $C_3 \approx 0.2$ and $C_4 \approx 1.8$.

These results are hallmarks of a transition from a bulk to an edge biofilm growth mode, triggered by nutrient limitation (*Pirt, 1967*). In the deceleration regime, diffusive transport of nutrients from a region external to the colony continues to sustain EPS production at the biofilm periphery, analogous to Stefan-like problems in solidification. Our generalized multiphase model is thus able to quantitatively rationalize the expansion curves, transition time and localized biological activity observed experimentally, and demonstrates that nutrient availability and diffusive transport governs the dynamics of *Bacillus subtilis* macrocolonies grown on agar.

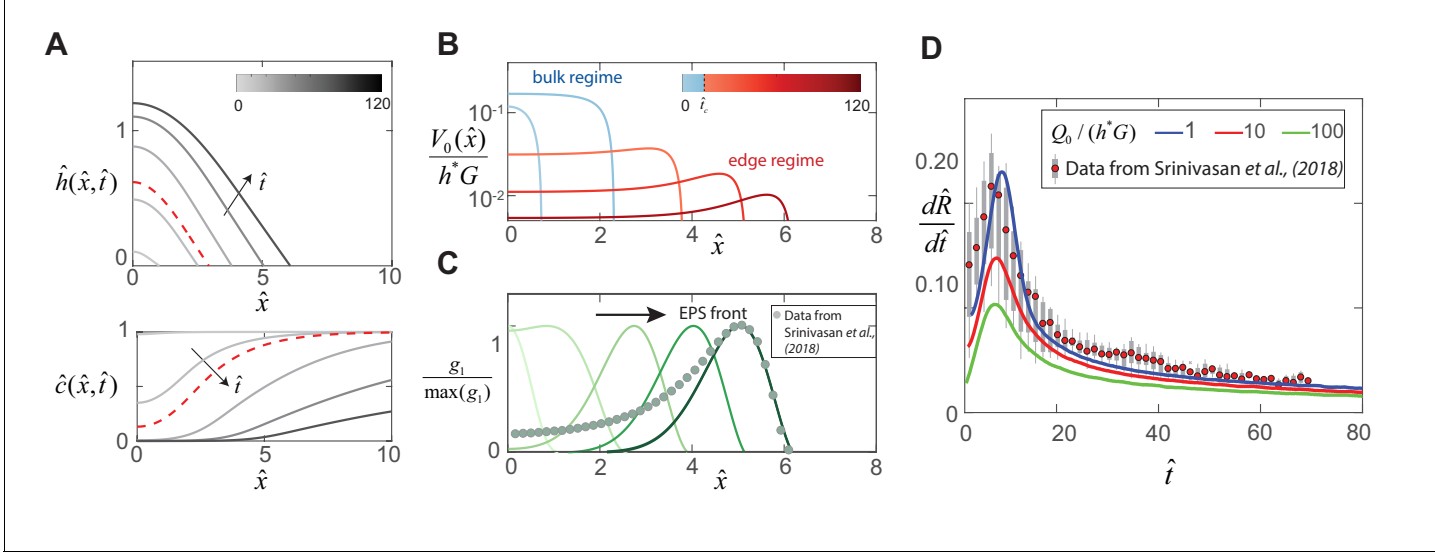

**Figure 4.** Dynamics of EPS production and biofilm expansion obtained by solving (12 - 14) with $\beta_1 = 6.7$, $\beta_2 = 0.01$, $\beta_3 = 0.02$, $\beta_4 = 1$ and $\beta_5 = 1.7$. (A) On the top are thickness profiles $\hat{h}(\hat{x}, \hat{t})$ of an expanding biofilm colony, at time intervals of $\hat{t} = 1, 5, 20, 50$ and $90$. The nutrient field $c(x, t)$ at corresponding time intervals is plotted at the bottom. The dotted red line indicates profiles at $\hat{t}_c = 8$, the transition point between the bulk and edge expansion regimes. (B) Variation of the vertical fluid uptake profile within the swarm calculated from *Equation (9)*. The light blue lines correspond to the bulk growth regime for $\hat{t} = 1, 5$ while the red lines correspond to $\hat{t} = 20, 50$ and $90$ in the edge growth regime. (C) Plots of normalized EPS production activity within the biofilm, where $g_1$ is evaluated using the expression in *Table 1*. The data points are spatial measurements of *tapA* gene activity in *B. subtilis* biofilms reproduced from *Srinivasan et al. (2018)*, with distances scaled by $L = 550$ µm. (D) Solid lines indicate transient colony edge expansion velocities for $\beta_4 = 1, 10$ and $100$ respectively, and with other parameter values fixed as listed above. The experimental data is reproduced from *Srinivasan et al. (2018)* and indicates median expansion velocities (filled circles), the 25th to 75th percentile velocities (filled box), and extreme values (vertical lines), where the data has been scaled by $U = 0.5\,\mu m s^{-1}$ and $G = 1/40$ min$^{-1}$.
DOI: https://doi.org/10.7554/eLife.42697.007

## Discussion

Analysis of collective microbial expansion in thin film geometries often prioritizes biological mechanisms, such as genetic regulation, developmental programs and cellular signaling/competition, over the role of the heterogeneous physical micro-environments. Here we have presented a multi-phase theory that quantitatively describes the expansion dynamics of microbial swarms and biofilms and considers variations in the colony thickness, an aspect of colony expansion that has often been overlooked in many theories (*Korolev et al., 2012*; *Ghosh et al., 2015*; *Wang et al., 2017*). The resulting unified description of both steady-state swarms and transient biofilm spreading leads to simple estimates and scaling laws for the colony expansion rate that are validated via comparison with experimental measurements for different systems. In swarms, exudation of water from the permeable substrate via bacterial osmolyte secretion facilitates steady state colony expansion. Numerical solutions of our model demonstrate that the shape of the swarm front is determined by capillarity, and its expansion speed by cell-division and growth, leading to scaling laws validated by comparison with previous experiments. In contrast, transient biofilm macrocolony expansion on agar is driven by osmotic polymer stresses generated via EPS matrix production in a spatially localized zone at the periphery. Nutrient transport and depletion leads to the formation of these heterogenous zones, and results in two regimes in biofilm expansion.

However our depth-integrated theory also has certain limitations. For example, we are unable to capture discrete thickness variations of the order of a few cells, which might require an agent-based approach. For bacterial swarms, our model is unable to quantitatively account for the region of enhanced thickness (i.e., the multilayer region in *Figure 1C and E*), likely because the multilayer width is difficult to experimentally ascertain, owing to the large tail distribution seen in the mean intensity trace in *Figure 1E*, and the arbitrariness in the choice of threshold in *Appendix 3—figure 5*. Similarly, in the context of biofilm colony expansion, our model does not account for sliding and

frictional contact between the cells/EPS matrix and the substrate (*Farrell et al., 2013*). More generally, our mean-field picture neglects fluctuation-driven effects during colony expansion, such as the formation multicellular raft structures (*Kearns, 2010*) and synchronized long-range interactions (*Chen et al., 2017*).

Natural next steps of our approach include (i) adding three-dimensional effects by allowing for spatial variations in the mechanical stresses, flows and nutrient fields in the vertical direction, (ii) accounting for orientational order in the bacterial swarms and films, and (iii) accounting for interfacial tension on the stability of the growing swarm/biofilm-fluid interface, especially in the context of fingering instabilities in microbial colonies *Trinschek et al. (2018)*.

A rigorous multi-phase approach may also be relevant in revisiting pattern formation phenomena in microbial colony expansion (*Matsushita et al., 1999*), that so far been addressed primarily using various non-linear diffusion models (*Golding et al., 1998*; *Allen and Waclaw, 2019*) that ignore the third dimension. Finally, from an experimental and theoretical perspective, our results naturally raise the question of controlling biofilm and swarm expansion by manipulating water and nutrient availability, complementing the better studied approaches of manipulating colonies by the genetic regulation of EPS production, cell division, and chemical signaling in microbial colonies.

## Acknowledgments

We thank M Cabeen, R Losick for providing strains, S Rubinstein for access to the fluorescent imaging and microbial culture facility, and H Berg, L Ping and A Pahlavan for helpful discussions. This work was supported by the Harvard MRSEC DMR 1420570 and the MacArthur Foundation (LM).

## Additional information

### Funding

| Funder | Grant reference number | Author |
| --- | --- | --- |
| National Science Foundation | DMR 1420570 | L Mahadevan |
| MacArthur Foundation | | L Mahadevan |

The funders had no role in study design, data collection and interpretation, or the decision to submit the work for publication.

### Author contributions

Siddarth Srinivasan, Conceptualization, Formal analysis, Software, Validation, Investigation, Methodology, Writing—original draft, Writing—review and editing; C Nadir Kaplan, Software, Formal analysis, Validation, Investigation, Writing—review and editing; L Mahadevan, Conceptualization, Formal analysis, Supervision, Funding acquisition, Validation, Investigation, Writing—original draft, Project administration, Writing—review and editing

### Author ORCIDs

L Mahadevan https://orcid.org/0000-0002-5114-0519

### Decision letter and Author response

Decision letter https://doi.org/10.7554/eLife.42697.022
Author response https://doi.org/10.7554/eLife.42697.023

## Additional files

### Supplementary files

• Source code 1. COMSOL file that implements the numerical solutions to *Equations (7)-(8)* governing bacterial swarm expansion.
DOI: https://doi.org/10.7554/eLife.42697.008

• Source code 2. COMSOL file that implements the numerical solutions to *Equations (12)-(14)* governing bacterial biofilm expansion.
DOI: https://doi.org/10.7554/eLife.42697.009

• Transparent reporting form
DOI: https://doi.org/10.7554/eLife.42697.010

### Data availability

All data generated or analysed are included in the manuscript.

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

# Appendix 1

DOI: https://doi.org/10.7554/eLife.42697.011

## Experiment

### Strains

In this study, we used two *B. subtilis* strains, MTC822 and MTC832, that were both previously constructed from a wild-type NCIB 3610 *B. subtilis* strain using a standard transformation protocol (*Sinha, 2013*). The MTC822 strain is used for fluorescence visualization in the swarming experiment, where the mkate2 red fluorescent protein reports on the activity of the constitutive hyperspank promotor via the *amyE::Phyperspank-mkate2* construct. The MTC832 strain was used in the biofilm experiments in order to visualize localized matrix production activity and harbors the *amyE::PtapA-cfp* construct. In the MTC832 strain, the cfp cyan fluorescent protein reports on the activity of the *tapA* gene that is associated with exopolysaccharide production activity.

### Materials and methods

Swarm plates were prepared using 0.5 wt% agarose gel (A1296, Sigma) infused with 25 ml of Luria-Bertani (Miller) medium (i.e. 10 g/L Tryptone 10 g/L NaCl 5 g/L Yeast Extract, Sigma) and 25 µg/ml Chloramphenicol. Biofilm plates were prepared using 1.5 wt% agarose gel (A1296, Sigma) infused with the standard MSgg biofilm-inducing growth medium (*Branda et al., 2001*) (i.e. 50 µM $MnCl_2$, 5 mM $KH_2PO_4$, 1 µM $ZnCl_2$, 50 µM $FeCl_3$, 2 mM $MgCl_2$, 700 µM $CaCl_2$, 50 µg/ml threonine, 50 µg/ml tryptophan, 50 µg/ml phenylalanine, 0.5 wt% glutamate, 0.5 wt% glycerol, 2 µM thiamine and 100 mM MOPS (pH 7)) and 50 µg/ml Spectinomycin. Note that all plates underwent an identical drying protocol prior to use. Freshly poured plates were initially dried with the lid open under a laminar flow hood for 15 min. Subsequently, the lid was closed and the dish was cooled at 25 C overnight for a period of 10 hr. All strains were initially grown in fresh Luria-Bertani (Miller) broth medium (Sigma) until mid-exponential phase in a shaker/incubator at 37C. The cultures were diluted to $OD_{650} = 0.1$ and $\sim 1$ $\mu l$ drop was deposited onto the corresponding swarm (for MTC822) or biofilm (for MTC832) plates. The petri-plates were transfered to a 30C incubator chamber during growth. Fluorescence imaging was performed using a Zeiss Axiozoom.V16 microscope with a PlanNeoFluar Z 1.0x objective (NA 0.25), with a Zeiss 63 HE filter to image the red mkate2 protein, and a Zeiss 47 HE filter to image the cyan cfp protein. For swarm profile measurements, images of the advancing swarm front were captured every 10 s over a period of 10 min. For biofilm colonies, expansion velocities were measured every 10 min over a period of 72 hr following the protocol described in *Srinivasan et al. (2018)*.

## Appendix 2

DOI: https://doi.org/10.7554/eLife.42697.011

### Theory

Consider a quasi-2D expanding swarm or biofilm colony where, $x$ denotes the horizontal direction and $z$ denotes the vertical (thickness) direction. The macroscopic colony is described by (i) the thickness field $h(x, t)$, (ii) the volume fraction of the active phase (i.e. swarmer cells or EPS matrix $\phi(x, t)$ and (iii) the averaged concentration field in the substrate $c(x, t)$.

### Nutrient transport

We assume that $\tilde{c}(x, z, t)$ is the nutrient concentration in the substrate denoted by region $-d < z < 0$, where $z = 0$ is the substrate/colony interface and $d$ is the substrate thickness. The time evolution of $\tilde{c}(x, z, t)$ is governed by the diffusion equation.

$$\frac{\partial \tilde{c}}{\partial t} - D\nabla^2\tilde{c} = 0 \tag{A1}$$

where $D$ is the diffusivity of the nutrient in the agarose gel. Integrating the diffusion equation through the substrate thickness results in,

$$\frac{\partial c}{\partial t} - D\frac{\partial^2 c}{\partial x^2} = -\frac{\Gamma h \phi}{d}\frac{c}{(K+c)} \tag{A2}$$

where $c(x, t) = (1/d)\int_{-d}^{0}\tilde{c}\,dz$ is the mean depth-averaged nutrient concentration in the substrate, $\Gamma$ is the specific nutrient consumption rate, and $K$ is the half velocity constant at which the specific growth rate is one half the maximum value. In obtaining **Equation (A2)**, we have balanced nutrient efflux from the substrate with nutrient consumption within the colony as $D(\partial\tilde{c}/\partial z)|_{-d} = -\Gamma h\phi c/(K+c)$. **Equation (A2)** describes nutrient transport within the permeable tissue/gel substrate underneath the colony, and holds when the diffusion time scale is much smaller than the flow time scale within the low-aspect-ratio substrate. Upon non-dimensionalizing **Equation (A2)**, the effective nutrient consumption rate $S$ used in **Table 1** in the main text is given by $S = \Gamma h^* \phi_{eq}/(c_0 d)$, where $h*$ is the colony thickness scale, $\phi_{eq}$ is the active phase volume fraction scale and $c_0$ is the initial nutrient concentration in the substrate.

**Appendix 2—table 1.** List of the symbols, descriptions, and numerical value for each of the parameters used.

| Variable | Description | Numerical value |
|---|---|---|
| $H$ | vertical length scale | swarms - 0.5 $\mu$m biofilms - 400 $\mu$m |
| $L$ | horizontal length scale | swarms - 100 $\mu$m biofilms - 1100 $\mu$m |
| $U$ | horizontal velocity scale | swarms - 1.3 $\mu$m/s biofilms - 0.5 $\mu$m/s |
| $\phi_0$ | equilibrium volume fraction of active phase | swarms - 0.5 biofilms - 0.04 |
| $Q_0$ | vertical fluid velocity scale | swarms - $10^{-2}$ $\mu$m/s biofilms - 0.04 |
| $g_0$ | effective swarm cell growth rate | 0.005 — 0.2 s$^{-1}$ |
| $G$ | EPS production rate | 1/40 min$^{-1}$ |
| $\mu_1$ | EPS matrix viscosity | $10^5$ Pa.s |
| $\mu_2$ | fluid viscosity | $10^{-3}$ Pa.s |
| $\zeta$ | friction coefficient per unit cell volume | $10^{-2}$ pN/($\mu$m s$^{-1}$) |

*Appendix 2—table 1 continued on next page*

*Appendix 2—table 1 continued*

| Variable | Description | Numerical value |
|---|---|---|
| $\gamma$ | fluid surface tension | $10^{-2}$ N/m |
| $\Psi_0$ | osmotic pressure scale | 2100 Pa |
| | Dimensionless parameters: Bacterial swarm | |
| $\alpha_1$ | $\dfrac{\gamma H}{\zeta L^3 U}$ | 0.2 |
| $\alpha_2$ | $\dfrac{Q_0}{H g_0}$ | 0.1 — 4.3 |
| | Dimensionless parameters: Bacterial biofilm | |
| $\beta_1$ | $\dfrac{\Psi_0 H^2}{\mu_1 U L}$ | 6.7 |
| $\beta_2$ | $\dfrac{\gamma H^3}{\mu_1 U L^3}$ | 0.01 |
| $\beta_3$ | $\dfrac{\gamma H}{\zeta \mu_1 L^3}$ | 0.02 |
| $\beta_4$ | $\dfrac{Q_0}{HG}$ | 1 |
| $\beta_5$ | $\dfrac{S}{G}$ | 2 |

DOI: https://doi.org/10.7554/eLife.42697.013

## Continuity

For a volume element within the microbial colony, the averaged conservation of mass (*Ishii and Hibiki, 2011*) for the active phase and the aqueous phase is expressed as,

$$\frac{\partial \phi}{\partial t} + \nabla \cdot (\phi \boldsymbol{u_1}) = g_1(h, \phi, c), \tag{A3}$$

$$\frac{\partial (1-\phi)}{\partial t} + \nabla \cdot ((1-\phi)\boldsymbol{u_2}) = 0, \tag{A4}$$

where $\phi$ is the volume fraction of the active phase (i.e. swarmer cells or biomass), $\boldsymbol{u_1} = (u_1, w_1)$ is the averaged velocity of the active phase, $\boldsymbol{u_2} = (u_2, w_2)$ is the averaged velocity of the fluid phase along the $x$ (horizontal) and $z$ (vertical) directions respectively. Integrating *Equations (A3) and (A4)* through the colony thickness leads to,

$$h \frac{\partial \phi}{\partial t} + \int_0^h \frac{\partial}{\partial x}(\phi u_1) + (\phi w_1)|_0^h = h g_1(h, \phi, c), \tag{A5}$$

$$h \frac{\partial (1-\phi)}{\partial t} + \int_0^h \frac{\partial}{\partial x}((1-\phi)u_2) + ((1-\phi)w_2)|_0^h = 0. \tag{A6}$$

The boundary conditions in *Equations (A5) and (A6)* for the vertical velocities $w_1$ and $w_2$ are,

$$w_1|_{z=0} = 0 \quad \text{and} \quad w_1|_{z=h} = \frac{\partial h}{\partial t} + u_1 \frac{\partial h}{\partial x}, \tag{A7}$$

$$w_2|_{z=0} = V_0(x) \quad \text{and} \quad w_2|_{z=h} = \frac{\partial h}{\partial t} + u_2 \frac{\partial h}{\partial x}, \tag{A8}$$

where the no-flux condition is applied to active phase at $z = 0$. For the aqueous phase, at $z = 0$, there is a spatial fluid influx described by $V_0(x)$ due to the osmotic pressure difference between the substrate and the microbial colony. Both phases at the upper interface at $z = h$ obey the kinematic boundary condition. In **Equations (A2), (A5) and (A6)**, the five unknown fields are $h$, $\phi$, $c$, $u_1$ and $u_2$. The remaining two equations for closure are obtained from an averaged momentum balance.

## Momentum balance

The averaged momentum balances for each phase can be expressed as (**Ishii and Hibiki, 2011**; **Drew, 1983**),

$$\nabla \cdot (\phi \sigma_1) + M = 0, \tag{A9}$$

$$\nabla \cdot ((1 - \phi)\sigma_2) - M = 0, \tag{A10}$$

where $\sigma_1$ is the averaged stress tensor acting on a volume element of the swarm cell/EPS matrix phase, $\sigma_2$ is the averaged stress tensor in a volume element of the fluid phase and $M$ denotes the total momentum transfer between the swarm cell/EPS matrix and the fluid phase. The interfacial momentum transfer term is expressed as (**Ishii and Hibiki, 2011**; **Drew, 1983**),

$$M = p_f \nabla \phi - \zeta \phi(u_1 - u_2). \tag{A11}$$

The first term in **Equation (A11)** denotes the force $p_f \nabla \phi$ due to the averaged interfacial pressure (**Drew, 1983**; **Ishii and Hibiki, 2011**) on the cells/EPS matrix by the surrounding fluid. Note that the momentum transfer terms in **Equations (A9)-(A10)** balance each other as the averaged pressures at the two-phase interface are equal, leading to a vanishing net buoyancy. The second term corresponds to the net viscous stokes drag, where $\zeta$ is a friction coefficient. While **Equations (A9–A11)** are generally applicable in describing both swarms and films, particular expressions that describe $\sigma_1, \sigma_2$ and $\zeta$ are unique to swarming and biofilm expansion and are discussed below for each case.

## Bacterial swarms

Within microbial swarms, the fluid phase is modeled as a Newtonian liquid with viscosity $\mu_2$ whereas the swarmer cells are treated as inviscid with an isotropic stress equal to the surrounding fluid pressure $p_f$. The averaged constitutive laws for the swarmer cell phase and the fluid phase are expressed as,

$$\sigma_1 = -p_f I, \tag{A12}$$

$$\sigma_2 = -p_f I + \mu_2 (\nabla u_2 + \nabla u_2^T). \tag{A13}$$

Substituting **Equations (A12) and (A13)** in **Equations (A9–A11)**, we obtain

$$-\phi \nabla p_f - \zeta \phi(u_1 - u_2) = 0, \tag{A14}$$

$$\nabla \cdot ((1 - \phi)\tau_2) - (1 - \phi)\nabla p_f + \zeta \phi(u_1 - u_2) = 0, \tag{A15}$$

where $\tau_2 = \mu_2(\nabla u_2 + \nabla u_2^T)$ is the deviatoric stress tensor. In bacterial swarms, the friction coefficient is $\zeta = \zeta_c/V_c \approx 18\mu_2/a^2$ where $a$ is the diameter of the cell, $\zeta_c \approx 3\pi\mu_2 a$ is the friction coefficient of a single swarmer cell from Stokes's law, $V_c = \pi a^3/6$ is the cell volume.

## Thin-film lubrication limit for swarms

We consider the limit of $h \ll R$, where $R$ is the colony radius. In the thin-film lubrication limit, combining **Equations (A14) and (A15)** results in the equation governing the mean horizontal fluid velocity to leading order,

$$\mu_2 \frac{\partial}{\partial z}\left((1-\phi)\frac{\partial u_2}{\partial z}\right) = \frac{dp_f}{dx}, \tag{A16}$$

where the fluid pressure is assumed to vary only in the horizontal direction, and is set by the local curvature of the swarm colony and the fluid surface tension as,

$$p_f = p_0 - \gamma \frac{\partial^2 h}{\partial x^2}, \tag{A17}$$

where $p_0$ is a constant (atmospheric) pressure. Integrating **Equation (A16)** twice and using the boundary conditions $u_2(0) = 0$ and $(du_2/dz)_{z=h} = 0$ lead to the expression for the averaged horizontal fluid phase velocity profile as,

$$u_2 = \frac{1}{(1-\phi)\mu_2}\frac{dp_f}{dx}\left(\frac{z^2}{2} - hz\right). \tag{A18}$$

The mean horizontal swarmer cell velocity is determined from the Darcy-type equation in **Equation (A14)** as,

$$u_1 = u_2 - \frac{1}{\zeta}\frac{dp_f}{dx}. \tag{A19}$$

## Growth rate of bacterial swarms

We note that bacterial swarming is typically associated with nutrient rich environments, where the initial concentration level $c_0 \gg \Gamma$, the specific nutrient consumption rate. Paradoxically, although the rate of bacterial cell division and volume expansion are exponential, the rate of swarm expansion is constant. Therefore, despite abundant nutrient availability, cell division must be halted at the swarm interior through non-nutrient meditated regulatory/signaling mechanisms, such that only a subpopulation of swarmer cells undergo cell division. In our study, we account for this effect by considering a simple logistic model for the growth term of the form,

$$g_1(h,\phi,c) = g_0\phi\left(1 - \frac{h\phi}{H\phi_0}\right), \tag{A20}$$

where $H$ is the swarm colony thickness at the interior, and $g_0$ is an effective growth rate that accounts for spatial localization in cell division. More specifically, if $L(x)$ describes the spatial profile of cell growth within a swarm colony, then $g_0 = \int_0^R L(x)dx / \int_0^R \phi[1 - (h\phi/(H\phi_0))]dx$, where $R$ is the radius of the swarm colony. Measurements of the spatial distribution of cell growth rates within the colony during swarming remain lacking. Consequently, in our model, we determine the value of $g_0$ by fitting it to the experimental data. We use a nonlinear least-squares solver to match steady-state expansion speeds obtained from solving **Equations (A21)-(A22)** to the experimental data for steady-state *B. subtilis* swarms (see **Figure 1D** in main text).

## Swarm equations

Combining **Equations (A5)-(A8)** with **Equations (A17)-(A19)** results in the dimensional thickness averaged equations for swarm colonies,

$$\frac{\partial}{\partial t}(h\phi) + \frac{\gamma}{3\mu_2}\frac{\partial}{\partial x}\left(h^3\frac{\phi}{1-\phi}\frac{\partial h^3}{\partial x^3}\right) + \frac{\gamma}{\zeta}\frac{\partial}{\partial x}\left(h\phi\frac{\partial^3 h}{\partial x^3}\right),$$
$$= g_0 h\phi\left(1 - \frac{h\phi}{H\phi_0}\right) \tag{A21}$$

$$\frac{\partial}{\partial t}(h(1-\phi)) + \frac{\gamma}{3\mu_2}\frac{\partial}{\partial x}\left(h^3\frac{\partial h^3}{\partial x^3}\right) = (1-\phi)V_0(x). \tag{A22}$$

The five boundary conditions are $(\partial h/\partial x)_{x=0} = (\partial h/\partial x)_{x\to\infty} = 0$, $(\partial^3 h/\partial x^3)_{x=0} = (\partial^3 h/\partial x^3)_{x\to\infty} = 0$, and $\phi(0) = \phi_0$, where $\phi_0$ is the volume fraction of the swarmer cells at the interior where there is no flow. As discussed in the main text, we impose the far-field boundary condition by choosing a domain much larger than the colony size.

## Bacterial films

In our model, the EPS matrix constitutes an active viscous hydrogel network with a polymer volume fraction $\phi(x,t)$, while the fluid phase is considered as a freely moving solvent of volume fraction $1 - \phi(x,t)$. The averaged constitutive laws for the EPS matrix phase and the fluid phase are expressed as,

$$\sigma_1 = -(\Psi + p_f)\mathbf{I} + \mu_1(\nabla\mathbf{u_1} + \nabla\mathbf{u_1^T}), \tag{A23}$$

$$\sigma_2 = -p_f\mathbf{I} + \mu_2(\nabla\mathbf{u_2} + \nabla\mathbf{u_2^T}), \tag{A24}$$

where $\mu_1$ is the viscosity of the EPS matrix, $\mu_2$ is the aqueous phase viscosity, $\Psi$ represents an effective swelling pressure in the biofilm EPS hydrogel and $p_f$ is the fluid pressure. Substituting *Equations (A23) and (A24)* in *Equations (A9–A11)*, we obtain

$$\nabla \cdot (\phi\tau_1) - \nabla(\phi\Psi) - \phi\nabla p_f - \zeta\phi(\mathbf{u_1} - \mathbf{u_2}) = 0 \tag{A25}$$

$$\nabla \cdot ((1-\phi)\tau_2) - (1-\phi)\nabla p_f + \zeta\phi(\mathbf{u_1} - \mathbf{u_2}) = 0, \tag{A26}$$

where $\tau_1 = \mu_1(\nabla\mathbf{u_1} + \nabla\mathbf{u_1^T})$ and $\tau_2 = \mu_2(\nabla\mathbf{u_2} + \nabla\mathbf{u_2^T})$ are the deviatoric stresses in the EPS phase and aqueous phase, respectively. In our model, we assume that the contribution of stress from the viscosity of the aqueous phase is negligible compared to the frictional drag due to flow between water and EPS polymer chain network. Consequently, *Equation (A26)* reduces to a Darcy-type law of the form,

$$\mathbf{u_1} - \mathbf{u_2} = \frac{1-\phi}{\zeta\phi}\nabla p_f, \tag{A27}$$

where $\kappa(\phi) = (1-\phi)/\phi$ is a volume fraction dependent permeability and the fricton coefficient is $\zeta = \mu_2/\xi^2$, provided that $\xi \sim 50$ nm is the polymer mesh network length scale. Substituting *Equation (A27)* in *Equation (A25)* results in,

$$\nabla \cdot (\phi\tau_1) = \nabla(\phi\Psi + p_f). \tag{A28}$$

## Thin-film lubrication limit for bacterial films

In the thin-film lubrication limit for biofilms when $h \ll R$, *Equation (A28)* reduces to

$$\mu_1\frac{\partial}{\partial z}\left(\phi\frac{\partial u_1}{\partial z}\right) = \frac{d\Pi}{dx}, \tag{A29}$$

where $\Pi = \phi\Psi + p_f$ is treated as an effective EPS phase pressure, as it is $d\Pi/dx$ that drives the viscous EPS flow. In *Equation (A29)*, the effective pressure is assumed to vary only in the

horizontal direction, and the fluid pressure is once again set by the curvature and fluid surface tension according to *Equation (A17)*. Moreover, the swelling pressure is expressed as $\Psi = \psi(\phi)/\phi$, where $\psi(\phi) = \psi_0 \times \phi^3$ is the osmotic pressure of the biofilm EPS matrix using the Flory-Huggins model for a polymer network in a $\theta$-solvent (*Cogan and Guy, 2010*; *Winstanley et al., 2011*). The osmotic pressure scale is $\psi_0 = kT/b^3$, where $b \sim 0.5$ nm is the approximate size of the monomer unit in the EPS matrix. Note that although we refer to $\Pi$ as an effective pressure, in *Equation (A23)* the mechanical pressure acting on the EPS phase is instead $\psi(\phi)/\phi + p_f$. Integrating *Equation (A29)* twice using the boundary conditions $u_1(0) = 0$ and $(du_1/dz)_{z=h} = 0$ results in,

$$u_1 = \frac{1}{\phi \mu_1} \frac{d\Pi}{dx} \left( \frac{z^2}{2} - hz \right).$$ (A30)

The mean horizontal fluid velocity is determined from the Darcy-type equation in *Equation (A27)* as,

$$u_2 = u_1 - \frac{1-\phi}{\zeta \phi} \frac{dp_f}{dx}.$$ (A31)

## Growth rate of bacterial films

In biofilms, the growth term $g_1(h, \phi, c)$ is expressed by the modified logistic term,

$$g_1(h, \phi, c) = G\phi \frac{c}{K+c} \left( 1 - \frac{h\phi}{H\phi_0} \right),$$ (A32)

where $G$ is the specific EPS production rate, $H$ is the maximum overall biofilm colony thickness and $\phi_0$ is the volume fraction of the EPS matrix in the fully swollen state. Note that, in contrast to microbial swarming, the growth rate term in expanding biofilms is strongly coupled to the local nutrient concentration field via a Monod-type term in *Equation (A32)*.

## Biofilm equations

Combining *Equations (A5)-(A8)* with *Equations (A30)-(A31)* results in the dimensional thin-film governing equations for biofilm colonies,

$$\frac{\partial}{\partial t}(h\phi) - \frac{1}{3\mu_1} \frac{\partial}{\partial x} \left( h^3 \frac{\partial \Pi}{\partial x} \right) = Gh\phi \frac{c}{K+c} \left( 1 - \frac{h\phi}{H\phi_0} \right),$$ (A33)

$$\frac{\partial}{\partial t}(h(1-\phi)) - \frac{1}{3\mu_1} \frac{\partial}{\partial x} \left( h^3 \frac{1-\phi}{\phi} \frac{\partial \Pi}{\partial x} \right)$$
$$+ \frac{\gamma}{\zeta} \frac{\partial}{\partial x} \left( h \frac{(1-\phi)^2}{\phi} \frac{\partial^3 h}{\partial x^3} \right) = (1-\phi) V_0(x),$$ (A34)

$$\frac{\partial c}{\partial t} - D \frac{\partial^2 c}{\partial x^2} = -\frac{\Gamma h\phi}{d} \frac{c}{K+c}.$$ (A35)

The eight boundary conditions are the symmetry conditions $(\partial h/\partial x)_{x=0} = (\partial h/\partial x)_{x=R} = 0$, $(\partial h^3/\partial x^3)_{x=0} = (\partial^3 h/\partial x^3)_{x=R} = 0$, $(\partial \phi/\partial x)_{x=0} = (\partial \phi/\partial x)_{x=R} = 0$, and $(\partial c/\partial x)_{x=0} = (\partial c/\partial x)_{x=R} = 0$ where $R$ is now the size of the petri dish that is infused with nutrients.

## Numerical computation

Numerical solutions to *Equations (A21)-(A22)* and *Equations (A33)-(A35)* were implemented using the COMSOL 5.0 finite element package and have been provided as source code files. We use a fixed 1D domain of size $x \in [0, L_1]$ where $L_1 = 16$ and 150 for the

biofilm and swarm simulations respectively. We use quintic Lagrange basis functions with element sizes below 0.04 and use the general form PDE solver. To handle the moving contact line, we introduce a precursor film of thickness $h_p/H = 0.0125$, where $H$ is the vertical length scale (see **Appendix 3—figure 1**). We follow the regularization described in **Trinschek et al. (2016)** to introduce a minimum threshold for growth and a stable fixed point in the precursor film. Specifically, the growth terms in (A2), (A20) and (A32) are multiplied by a factor $F = [1 - \exp\left(5\left(\hat{h}_p - \hat{h}\hat{\phi}\right)\right)] \times \left(1 - \hat{h}_p/(\hat{h}\hat{\phi})\right)$, where $F \approx 1$ everywhere except near the precursor film where $F = 0$.

## Experimental data

**Appendix 2—table 2.** Summary of the comparision between the experimental data and model. The experimentally measured quantities are the colony expansion speed $V = dR/dt$ and multilayer region thickness $W$. The value of $\alpha_2$ is determined by fitting **Equations (7)-(8)** to the expansion velocity, leading to estimates of the effective growth rate $g_0$, the horizontal length scale $L$ and the capillary number $Ca$.

| No | Reference | $V = \frac{dR}{dt}$ $\mu$ms$^{-1}$ (exp) | $W$ $\mu$m (exp) | $\alpha_2$ (model) | $g_0$ s$^{-1}$ (model) | $L$ $\mu$m (model) | $Ca$ $\times 10^{-7}$ (model) |
|----|-----------|------------|----------|---------|---------|---------|----------|
| 1 | *B. subtilis* this work - (data set I) | 0.56 | 178 | 1.49 | 0.013 | 98 | 1.32 |
| 2 | *B. subtilis* this work - (data set II) | 0.26 | 368 | 4.33 | 0.005 | 128 | 0.59 |
| 3 | *B. subtilis* this work - (data set III) | 0.60 | 132 | 1.35 | 0.015 | 96 | 1.42 |
| 4 | *E. coli* **Wu and Berg, 2012** | 1.7 | 154 | 0.30 | 0.07 | 66 | 4.38 |
| 5 | *E. coli* **Darnton et al., 2010** | 3.8 | - | | | | |

DOI: https://doi.org/10.7554/eLife.42697.014

## Appendix 3

DOI: https://doi.org/10.7554/eLife.42697.011

### Figures

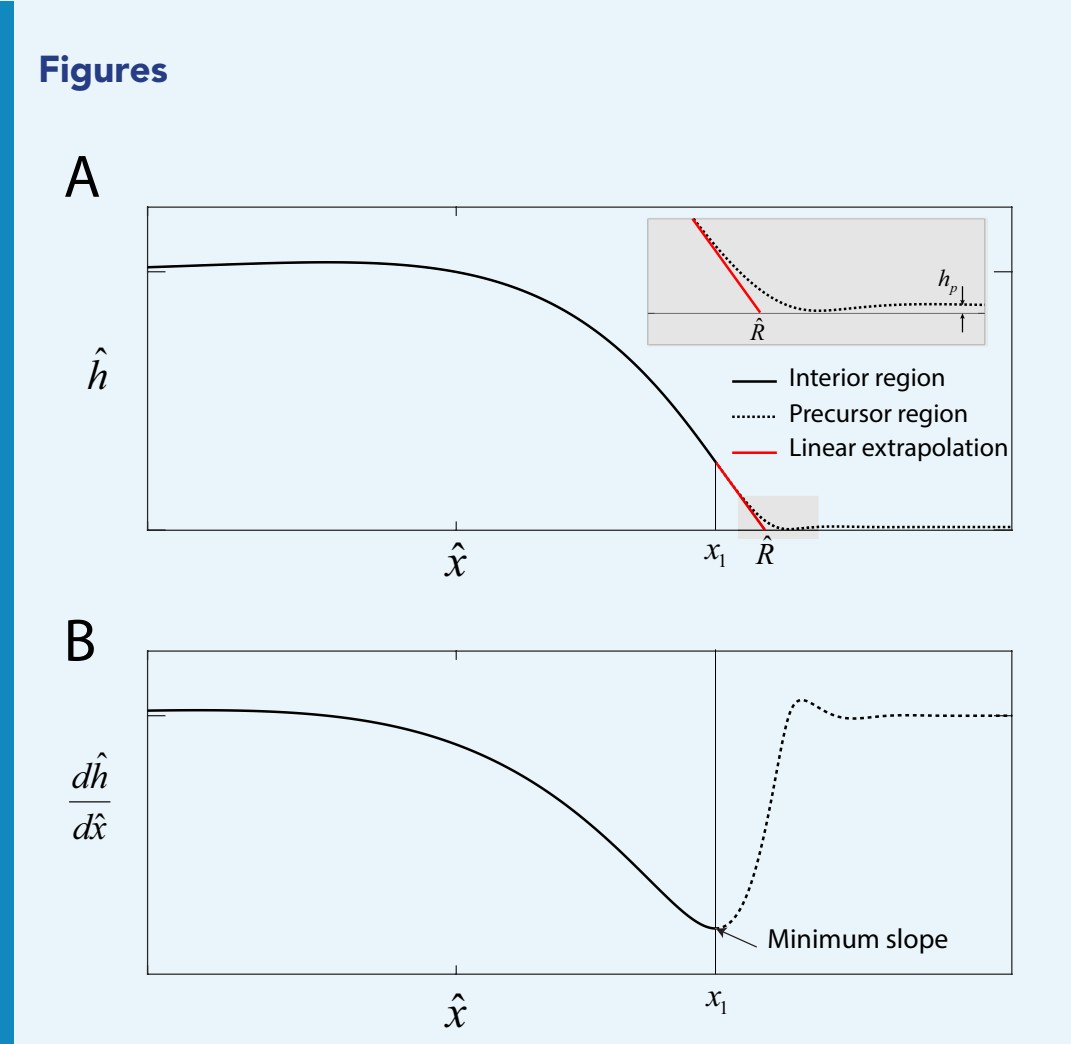

**Appendix 3—figure 1.** Precursor film. (**A**) Numerical solution to *Equations (5)-(6)* in the main text that represents the swarm profile $\hat{h}(\hat{x})$. The solid black line indicates the swarm profile in the interior region domain $\hat{x} \in [0, x_1]$. The dashed line beyond $x_1$ denotes the precursor-film region. The solid red line represents a linear extrapolation of the swarm profile in the region $\hat{x} \in [x1, \hat{R}]$, where $\hat{R}$ is the swarm radius. Inset: Magnified view of the transition region, where $h_p$ is the precursor film of thickness. (**B**) The onset of the precursor film is defined at the point of the numerical profile where the slope is a minimum.

DOI: https://doi.org/10.7554/eLife.42697.016

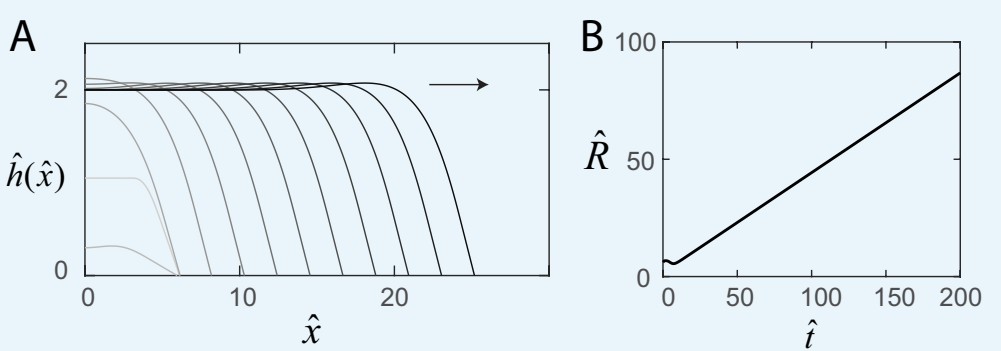

**Appendix 3—figure 2.** Steady-state swarm solutions. (**A**) The evolution of the numerical swarm thickness $\hat{h}(\hat{x}, \hat{t})$ plotted in the laboratory frame at fixed time intervals. (**B**) Plot of the swarm radius as a function of time indicating steady-state solutions.

DOI: https://doi.org/10.7554/eLife.42697.017

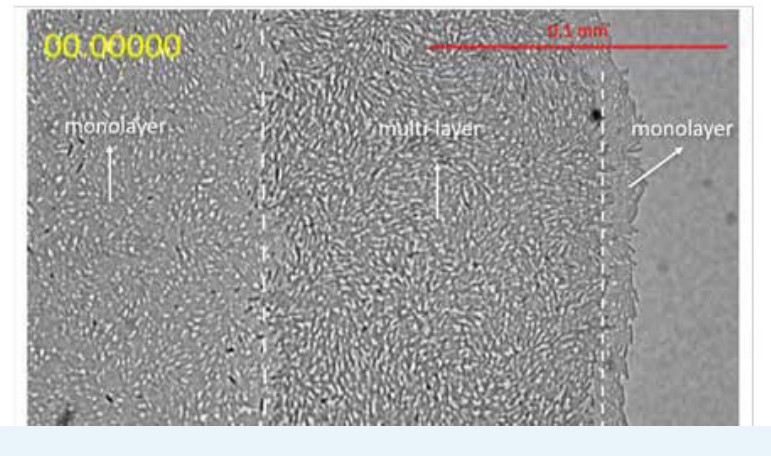

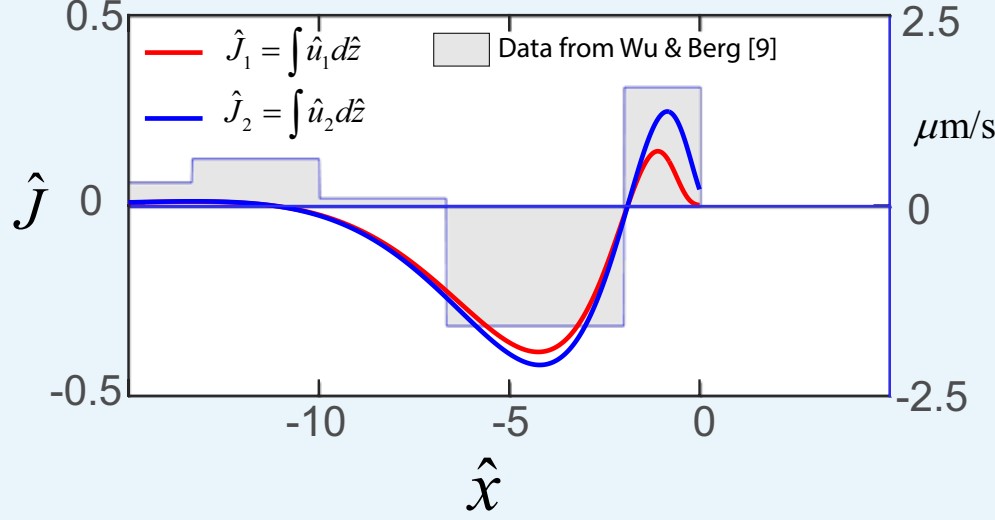

**Appendix 3—figure 3.** Top: Image of edge of swarm colony. *B. subtilis* swarm colony showing the interior monolayer region, the multi-layer front, and the monolayer at the very edge of the colony. Bottom: Horizontal flows in swarms. Steady-state profile of the dimensionless net horizontal velocity $\hat{J} = \int_0^{\hat{h}} \hat{u}\,d\hat{z}$ in the swarmer cell phase (red) and fluid phase (blue). Expression for $\hat{J}_1$ and $\hat{J}_2$ are obtained from **Equations (A18) and (A19)** as $J_1 = -\frac{1}{1-\phi}\frac{d\hat{p}}{d\hat{x}}\frac{\hat{h}^3}{3} - \alpha_1 \frac{d\hat{p}}{d\hat{x}}\hat{h}$ and

$J_2 = -\frac{1}{1-\phi}\frac{d\hat{p}}{d\hat{x}}\frac{\hat{h}^3}{3}$, where $\alpha_1 = 0.2$ as discussed in the main text. The step function represents experimental flow speeds as a function of the distance form the swarm edge as measured by **Wu and Berg (2012)**, and where the horizontal distance has been scaled by $L = 15\mu$m.

DOI: https://doi.org/10.7554/eLife.42697.018

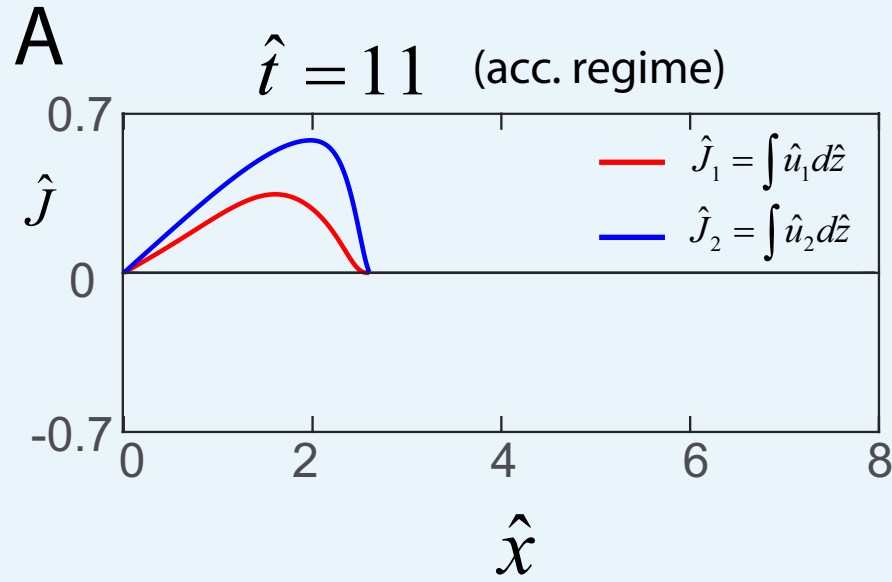

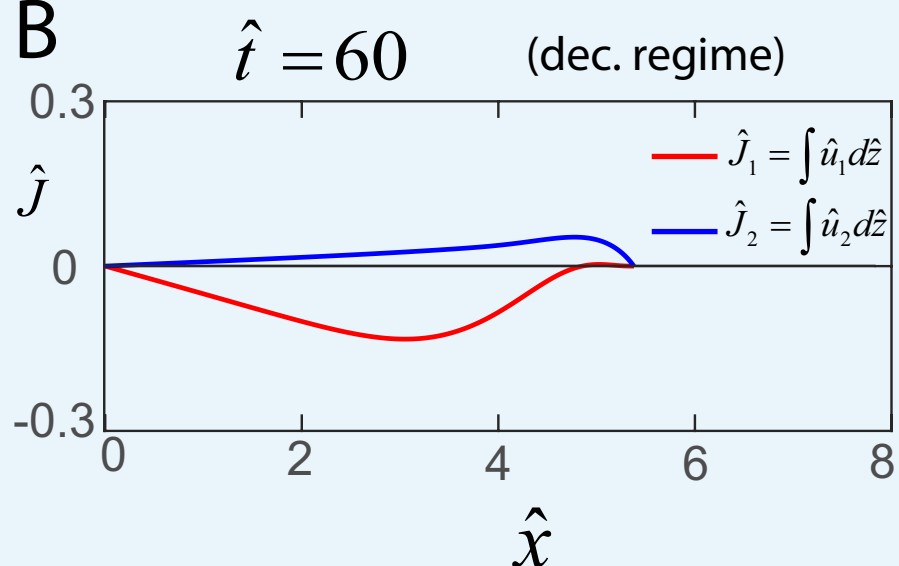

**Appendix 3—figure 4.** Early and late flows in biofilms. Profile of the dimensionless net horizontal velocity $\hat{J} = \int_0^{\hat{h}} \hat{u}d\hat{z}$ in the EPS phase (red) and fluid phase (blue). Expression for $\hat{J}_1$ and $\hat{J}_2$ are obtained from **Equations (A30) and (A31)** as $J_1 = -\frac{\beta_1}{\phi}\frac{d\hat{\Pi}}{d\hat{x}}\frac{\hat{h}^3}{3}$ and $J_2 = -\frac{\beta_1}{\phi}\frac{d\hat{\Pi}}{d\hat{x}}\frac{\hat{h}^3}{3} - \beta_3\frac{1-\phi}{\phi}\frac{d\hat{p}}{d\hat{x}}\hat{h}$, where $\beta_1 = 6.7$ and $\beta_3 = 0.02$ as discussed in the main text.

DOI: https://doi.org/10.7554/eLife.42697.019

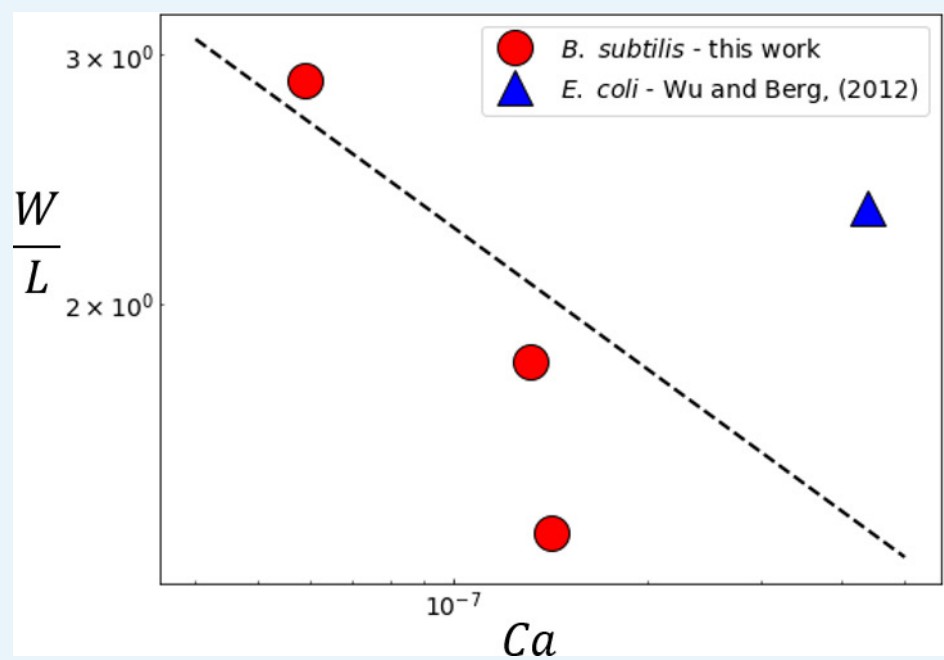

**Appendix 3—figure 5.** Variation of the experimentally measured multilayer swarm width $W$ with $Ca$. For *B. subtilis* swarm experiments, the value of $W$ was determined by considering the width of the region where the mean constitutive fluorophore intensity $I > 0.7\max(I)$ (See **Figure 1E**). The multilater width in the *E. coli* swarm was reported as 154 μm ± 27 μm by **Wu and Berg (2012)**. The dashed line corresponds to the predicted scaling from **Equation (6)**.
DOI: https://doi.org/10.7554/eLife.42697.020

