## [Decision Letter]

Thank you for submitting your article "Dynamics of spreading microbial swarms and films" for consideration by *eLife*. Your article has been reviewed by three peer reviewers, and the evaluation has been overseen by a Reviewing Editor and Arup Chakraborty as the Senior Editor. The following individual involved in review of your submission has agreed to reveal his identity: Alex Persat (Reviewer #1).

The reviewers have discussed the reviews with one another and the Reviewing Editor has drafted this decision to help you prepare a revised submission.

Summary:

Srinivasan and coauthors propose a generalized model to describe the expansion of bacterial communities growing on agar plates in two distinct modes: swarms and biofilms. They demonstrate that the interplay between biological mechanisms, such as osmolyte secretion, extracellular matrix production and spatial localization in cell growth, and physics of the microbial environment, such as hydrodynamics, mechanics, osmotic flux and nutrient diffusion, controls the expansion dynamics of microbial swarms and biofilms. They show that swarming colonies undergo steady state expansion, while biofilms expand transiently, going from an initial acceleration phase to a decelerating one. These observations are in agreement with a few experiments. In the end, they prove that despite the differences in biological mechanisms of films and swarms, a unified two-phase model of a spreading colony can describe both systems.

The reviewers found the paper well-written and the theoretical and numerical analysis was performed very carefully. The phase-field theory developed in the paper goes substantially beyond earlier models as recently reviewed by Allen and Waclaw, 2018. The detailed parameter list in the Appendix can become a valuable reference for future studies. The manuscript thus offers a significant theoretical advance.

Title:

The reviewers thought that the current title sounds like an area of research rather than the title of a paper. Perhaps adding the term "model" or "theory" or any other more specific/descriptive statements would more clearly reflect the content of the paper.

Essential revisions:

The reviewers found that the manuscript lacks a bit of clarity with respect to conclusions and agreement with experimental data:

1) There is a lack of comparison between scaling and experiments. I think the beauty of this work is to highlight scaling of the front velocity and width of the front. However, the authors limit their validation as fitting a single data point to obtain prefactors (e.g. subsection “Steady state swarms”, last paragraph). I would expect that the authors would make an effort to measure velocities for a few capillary numbers for example, and verify the -1/3 exponent behavior. It looks like the authors can actually do these experiments since they already have some data shown in Figure 1. This would absolutely strengthen the paper and the model (independently of whether the model fits the data).

2) The aim of the model is to describe both swarming and biofilm expansion and weigh in on their physical differences. However, describing swarming and biofilm separately makes it difficult to have a direct comparison between the two. It would be useful to directly compare the two (in concomitance with Table 1), explaining, for example, the choices of different active-passive phases, osmotic flux, growth term and nutrient uptake. For example, it would be useful to know why in swarmer cells nutrient uptake is not considered (is it related to nutrient diffusion in different agar substrate?). Why isn't surface tension considered in the biofilm model? This lack of discussion weakens the argument of a unifying model.

3) In biofilms, it is not clear to me why the active phase consists of only the EPS and cell growth is not considered as a driver of expansion.

4) I would imagine that one important factor limiting growth particularly in the biofilm case is the friction between the colony and the agar substrate. This is pointed out in Farell, 2013, but it seems to be vastly overlooked by the authors. Can they incorporate this in their fluidic model?

5) They state that their model reveals that bacterial swarm colonization corresponds to a fluid regulated limit, whereas the dynamics of biofilm expansion is governed by nutrient transport (subsection “Conclusions”). However, these are actually the assumptions used to describe the system, so it is not so surprising. Can the authors comment on this point?

One reviewer identified himself as "eviewer A" of the previous submission to PNAS for this manuscript (i.e. the positive reviewer), and stated that the authors had responded to his comments in their "response to reviewers" document, which they attached to the *eLife* submission.

This reviewer did note the following:

6) It is still slightly worrisome that 2 out of the 4 figures are introductory/motivational, as pointed out in my major comment 1 of the PNAS report.

7) Regarding my previous comment 2, about limitations of the model. I appreciate that the authors have added the sentence about shortcomings of the model after I pointed out one issue. Given that *eLife* does not have any length restrictions, I think it would be appropriate to add a new section to the main text where general limitations of the model are discussed, and where the authors believe the model is failing. I think such a section would strengthen the manuscript, provide additional context, and it would be helpful to the readers.

---

## [Author Response]

As per the reviewers’ recommendation, we have changed the title to “A multiphase theory for spreading microbial swarms and films”.

Essential revisions:1) There is a lack of comparison between scaling and experiments. I think the beauty of this work is to highlight scaling of the front velocity and width of the front. However, the authors limit their validation as fitting a single data point to obtain prefactors (e.g. subsection “Steady state swarms”, last paragraph). I would expect that the authors would make an effort to measure velocities for a few capillary numbers for example, and verify the -1/3 exponent behavior. It looks like the authors can actually do these experiments since they already have some data shown in Figure 1. This would absolutely strengthen the paper and the model (independently of whether the model fits the data).

We thank the reviewers for their comment. Following the reviewers’ suggestion, we have now included additional comparisons with experiments that verify some of the predictions of our model. Specifically, we now fit our model and compare the scaling behavior using five independent experimental measurements of swarm expansion velocities. These include two new experimental measurements of swarm expansion in *B. subtilis*, and two existing measurements in *E. coli* previously reported by Darton et al., 2010, and Wu and Berg, 2012. The comparison between the various experiments and fitting to our model is described in the main text in Appendix 2—table 2 and Figure 3D.

As seen in the figure for the velocity scaling, the measured swarm expansion velocities (dR/dt) indeed follows a -1/3 exponent with the Capillary number. We have introduced a new figure in the main text (i.e., Figure 3D) to emphasize the verification of the -1/3 scaling. We have rewritten the introduction of the time and length scales, that clarifies the estimation of the various parameters.

“There are two key dimensionless parameters that describes swarm colony expansion. […] A complete set of parameters for three experimental measurements of swarm expansion in *B. subtilis*, and two existing measurements in *E. coli* previously reported by Darton et al., 2010, and Wu and Berg, 2012, are summarized in Appendix [3].”

Finally, upon considering the new dataset, the width of the swarm front does not follow the scaling we had originally proposed, as seen in Appendix 3—figure A5.

For each experiment, we fit our model using the effective growth rate g0 as the fitting parameter. We have also included a discussion of this fitting, and the overall comparison between model and theory with the new experimental measurements:

“Furthermore, we corroborate our scaling law in Equation 5 by fitting our model to five independent experimental measurements of swarm expansion velocities, as shown in Figure 3D. [...] From an experimental point of view, the width of the multilayer region is not sharply defined in Figure 1E, and will depend on the choice of threshold.”

As the reviewer indicates, we think these results strengthen the paper and our model, especially the agreement of the scaling law with the experimental velocities. We are once again grateful and thank the reviewer for their valuable insight towards improving the manuscript.

2) The aim of the model is to describe both swarming and biofilm expansion and weigh in on their physical differences. However, describing swarming and biofilm separately makes it difficult to have a direct comparison between the two. It would be useful to directly compare the two (in concomitance with Table 1), explaining, for example, the choices of different active-passive phases, osmotic flux, growth term and nutrient uptake. For example, it would be useful to know why in swarmer cells nutrient uptake is not considered (is it related to nutrient diffusion in different agar substrate?). Why isn't surface tension considered in the biofilm model? This lack of discussion weakens the argument of a unifying model.

We thank the reviewers for their comment. As per the reviewers’ suggestion, we have now added a number of paragraphs (immediately after Table 1), that directly compares the various terms in our unifying model across swarms and biofilms. We have now described the origin of the differences in the nutrient uptake, growth term and the active and passive fluxes in parallel with Table 1.

Our Discussion directly addresses some of the questions the reviewers raise, viz.

1) The overall rate of change of nutrient concentration in the substrate depends on nutrient diffusion and uptake as described by Appendix 2—equation A2, i.e.

∂c∂t-D∂2c∂x2=-ΓhϕdcK+c

where, c(x,t) is the mean depth-averaged nutrient concentration in the substrate, Γis the specific nutrient consumption rate per unit concentration, k is the half velocity constant at which the specific growth rate is one half the maximum value and d is the substrate thickness. When the substrate concentration is scaled by the initial concentrationc0, the depletion term on the RHS of Equation 18 depends on Γ/c0, the ratio of the specific nutrient consumption rate to the initial concentration.

Bacterial swarming is typically associated with nutrient rich conditions, wherec0≫Γ. As a result, the nutrient uptake term can be neglected in bacterial swarming asg2→0, and the concentration c≈c0 throughout swarm expansion. In contrast, biofilm growth occurs under nutrient limited conditions whereΓc0∼O1, resulting in a corresponding uptake term as shown in Table 1.

2) Surface tension is indeed included in the biofilm model, and sets the capillary fluid pressure aspf∼-γ∂2h/∂x2. Furthermore, as shown in Appendix 2—equations A28 and A2944-45, the gradient of the effective EPS phase pressure Π drives the horizontal fluxes Q1(x) and Q2(x) of the EPS matrix and fluid phase in Table 1. Note that Π is a sum of the Flory-Huggins osmotic pressure and the capillary fluid pressure. In this manner, surface tension is indeed considered in the biofilm model.

3) In biofilms, it is not clear to me why the active phase consists of only the EPS and cell growth is not considered as a driver of expansion.

Our model considers the growth of mature, macroscopic biofilm colonies. In these colonies, the biomass almost entirely consists of the EPS matrix. For example, as stated in Flemming and Wingender, 2010.

“In most biofilms, the microorganisms account for less than 10% of the dry mass, whereas the matrix can account for over 90%. The matrix is the extracellular material, mostly produced by the organisms themselves, in which the biofilm cells are embedded.”

Moreover, the total biomass itself constitutes only ~5% of the biofilm by mass (with the fluid being the predominant phase). Therefore, we treat the biofilm as a highly viscous and hydrated gel. Our model considers the osmotic stress and swelling pressure in freshly produced EPS matrix as the major contributor to biofilm expansion.

It only at the very early stages of biofilm colony growth (when EPS production is minimal) where cell-cell contact is the major contributor to growth pressure. An example of the latter scenario is microcolonies that consist of only ~100 – 1000s of cells. Models of microcolonies do indeed consider stress contributions from growth, for example, as in the work of Farrell, 2013.

4) I would imagine that one important factor limiting growth particularly in the biofilm case is the friction between the colony and the agar substrate. This is pointed out in Farell, 2013, but it seems to be vastly overlooked by the authors. Can they incorporate this in their fluidic model?

In our model, we consider the both the active EPS matrix phase, and the passive aqueous phase as Newtonian fluids. Consequently, friction between the colony and the agar substrate is accounted for in our model by the no-slip boundary condition, u1z=0=0 and u2z=0=0, where u1 and u2 are the horizontal components of the EPS phase and fluid phase velocities.

Therefore, our assumption of a no-slip boundary condition corresponds to the limit of an infinite friction coefficient in the discrete cell 2D model of Farrell, 2013. There are two key factors that differentiates our model from that of Farrell, 2013: (i) the presence of the EPS matrix, and (ii) the consideration of a third (i.e., vertical thickness) dimension.

Variations in the biofilm thickness allow the EPS matrix to sustain shear gradients in velocity. As a result, the biofilm colony “flows” over the agar substrate as a fluid in our model, while individual bacterial cells expand by “sliding” as a 2D dimensional colony by sliding in the model of Farrell et al. Similarly, in our model EPS production and growth in the third dimension generates the osmotic stresses and swelling pressure that drives colony expansion.

Note also that Farrell et al. consider the very early micro-colony stage of biofilms (< 2hrs), where cells are present as a monolayer and both EPS production and variations in the third dimension are absent.

One possibility towards incorporating the effect of frictional sliding in macroscopic biofilms would be the introduction of a Navier slip-length parameter in our model, where the slip-length is tuned to account for sliding of EPS matrix on the agar substrate. We have added this point to the Discussion, where we discuss limitations and scope of our model, by the following line:

“Similarly, during biofilm colony expansion, we do not account for sliding and frictional contact between the cells/EPS matrix and the substrate (Farrell, 2013).”

5) They state that their model reveals that bacterial swarm colonization corresponds to a fluid regulated limit, whereas the dynamics of biofilm expansion is governed by nutrient transport (subsection “Conclusions”). However, these are actually the assumptions used to describe the system, so it is not so surprising. Can the authors comment on this point?

We thank the reviewer for this observation. We agree, our model does not reveal this statement and is indeed an assumption used to describe the system. We have rephrased the sentence to emphasize that our model merely “rationalizes the assumption”, rather than reveals the statement.

One reviewer identified himself as "reviewer A" of the previous submission to PNAS for this manuscript (i.e. the positive reviewer), and stated that the authors had responded to his comments in their "response to reviewers" document, which they attached to the eLife submission.This reviewer did note the following:6) It is still slightly worrisome that 2 out of the 4 figures are introductory/motivational, as pointed out in my major comment 1 of the PNAS report.

We thank the reviewer for the comment. We have now included an additional subpanel in Figure 3D to show the scaling of the velocity with the capillary number. We also note that despite having 2 figures, each figure has a number of sub-panels that could easily be expanded as separate figures. In the interest of brevity and condensing the results of the data and model in one location for both swarms and biofilms, we have opted to use the subfigure approach, rather than expanding each panel into separate figures.

7) Regarding my previous comment 2, about limitations of the model. I appreciate that the authors have added the sentence about shortcomings of the model after I pointed out one issue. Given that eLife does not have any length restrictions, I think it would be appropriate to add a new section to the main text where general limitations of the model are discussed, and where the authors believe the model is failing. I think such a section would strengthen the manuscript, provide additional context, and it would be helpful to the readers.

As the reviewer has suggested, we have now added paragraphs under the conclusions section where we discuss the limitations of our model and scope for expansion:

“Analysis of collective microbial expansion in thin film geometries often prioritizes biological mechanisms, such as genetic regulation, developmental programs and cellular signaling/competition, over the role of the heterogeneous physical micro-environments. […] Finally, from an experimental and theoretical perspective, our results naturally raise the question of controlling biofilm and swarm expansion by manipulating water and nutrient availability, complementing the better studied approaches of manipulating colonies by the genetic regulation of EPS production, cell division, and chemical signaling in microbial colonies.”